# Evolutionary dynamics of whole-body regeneration across planarian flatworms

Miquel Vila-Farré [1,2] ✉, Andrei Rozanski [1,2,12], Mario Ivanković [1,2,12], James Cleland[1,2,12], Jeremias N. Brand[1], Felicia Sandberg[3,4], Markus A. Grohme [2], Stephanie von Kannen[2], Alexandra L. Grosbusch[5], Hanh T.-K. Vu[1,2,6], Carlos E. Prieto[7], Fernando Carbayo[8], Bernhard Egger [5], Christoph Bleidorn[3], John E. J. Rasko [9,10,11] & Jochen C. Rink [1,2] ✉

Regenerative abilities vary dramatically across animals. Even amongst planarian flatworms, well-known for complete regeneration from tiny body fragments, some species have restricted regeneration abilities while others are almost entirely regeneration incompetent. Here, we assemble a diverse live collection of 40 planarian species to probe the evolution of head regeneration in the group. Combining quantification of species-specific head-regeneration abilities with a comprehensive transcriptome-based phylogeny reconstruction, we show multiple independent transitions between robust whole-body regeneration and restricted regeneration in freshwater species. RNA-mediated genetic interference inhibition of canonical Wnt signalling in RNA-mediated genetic interference-sensitive species bypassed all head-regeneration defects, suggesting that the Wnt pathway is linked to the emergence of planarian regeneration defects. Our finding that Wnt signalling has multiple roles in the reproductive system of the model species *Schmidtea mediterranea* raises the possibility that a trade-off between egg-laying, asexual reproduction by fission/regeneration and Wnt signalling drives regenerative trait evolution. Although quantitative comparisons of Wnt signalling levels, yolk content and reproductive strategy across our species collection remained inconclusive, they revealed divergent Wnt signalling roles in the reproductive system of planarians. Altogether, our study establishes planarians as a model taxon for comparative regeneration research and presents a framework for the mechanistic evolution of regenerative abilities.

Regeneration, defined here as the reformation of body parts lost to injury, is widespread in the animal kingdom[1]. Yet regenerative abilities often vary dramatically between closely related groups or species[2–6]. Between the likely ancestral nature of whole-body regeneration in animals[7] and the seemingly adaptive effect of regenerative abilities, a fascinating question arises: why is it that only some, but not all, animals are capable of regeneration?

Probing the reasons for the variation of regeneration is a challenge because such attempts necessarily need to link the molecular and cellular basis of regeneration (proximate mechanisms) to natural selection (ultimate mechanisms)[8]. As a taxon, planarian flatworms (Platyhelminthes, Tricladida) are especially suitable for such a systematic examination. On the one hand, many planarian species can regenerate complete animals from arbitrary tissue fragments. *Schmidtea*

**Fig. 1 | The MPI-NAT planarian collection. a**, Sampling sites (black dots) and collection location (red). **b**–**d**, Live images of flatworms that were characterized as part of this study. **b**, Planarians order Tricladida, Suborder Continenticola. From left to right. First row, *Bdellocephala angarensis*; *Bdellocephala* cf. *brunnea*; *Dendrocoelum lacteum*; *Crenobia alpina*; *Polycelis felina*; *Polycelis tenuis*; *Polycelis nigra*; *Seidlia* sp.; cf. *Atrioplanaria*. Second row, *Phagocata gracilis*; *Hymanella retenuova*; *Phagocata pyrenaica*; *Planaria torva*; *Cura pinguis*; *Cura foremanii*; *Schmidtea lugubris*; *Schmidtea polychroa* (dark strain); *Schmidtea polychroa* (unpigmented strain). Third row, *Schmidtea nova*; *Smed* (asexual strain); *Dugesia tahitiensis*; *Dugesia sicula*; *Dugesia* sp.; *Dugesia japonica*; *Girardia dorotocephala*; *Girardia tigrina*; *Spathula* sp. 3. **c**, Planarians order Tricladida, Suborder Maricola: *Camerata robusta*; *Procerodes littoralis*; *Procerodes plebeius*; *Bdelloura candida*; *Cercyra hastata*. **d**, Order Prolecithophora: *Plagiostomum girardi*; *Cylindrostoma* sp. Scale bar, 1 mm unless otherwise noted; measured (pixel resolution; solid line) or approximated during live imaging (graph paper; dotted line).

*mediterranea* (*Smed*) and *Dugesia japonica* have been developed into molecularly tractable model species[9–11]. Insights into the mechanistic underpinnings of planarian whole-body regeneration include the critical importance of their abundant pluripotent stem cells (neoblasts) as sole division-competent cells in somatic tissues[12,13] and a collectively self-organizing network of positional identity signals that orchestrate neoblast cell fate choices[14,15]. A key signal in patterning the anteroposterior axis (A–P axis) is the evolutionarily conserved Wnt signalling pathway[16–19], which signals via inhibiting the constitutive degradation of the transcriptional regulator ß-CATENIN-1 and via the ensuing changes to gene expression[20]. The upregulation of Wnt signalling at a planarian wound site is necessary and sufficient for tail specification[18], whereas Wnt inhibition is necessary and sufficient for head specification[17,19,21].

On the other hand, other planarian species have more limited regenerative abilities[22–24]. Interestingly, the experimental inhibition of canonical Wnt signalling is sufficient for rescuing head regeneration in

three planarian species with restricted (regionally limited) head regeneration[25–27], indicating that the misregulation of this signalling pathway contributes to the examined regeneration defects. With hundreds of planarian species in existence worldwide[28], a documented range of regenerative abilities from whole-body regeneration to the complete absence of regeneration[23] and emerging insights into the mechanistic underpinnings of regeneration from the molecularly tractable model species, planarians provide a compelling opportunity for investigating the evolution of regeneration.

Here we systematically explore the gain and loss of head regeneration across the planarian phylogeny via a live collection of more than 40 planarian species. Our analysis provides a first systematic overview of trait evolution within the taxon, identifies multiple independent transitions in head-regeneration ability and establishes the general Wnt-dependence of planarian regeneration defects. Our demonstration of positive Wnt signalling pleiotropies in the reproductive system of Smed makes a trade-off between egg-laying and asexual reproduction by fission/regeneration a plausible driver of regenerative trait evolution in planarians. Overall, our study highlights the utility of planarians as model taxon for the evolution of regeneration and provides a framework for analysing the underlying proximate and ultimate mechanisms.

## Results

### Comparative analysis of planarian head regeneration

Towards our goal of establishing planarians (Tricladida) as model taxon for the evolutionary dynamics of regeneration, we carried out worldwide field collections in locations with high planarian species diversity[29–33] (Fig. 1a and Supplementary Fig. 1a; Methods). We mainly focused on freshwater species due to the known variations in regenerative abilities[23,34] and broad compatibility with established planarian husbandry protocols[11,35,36]. Field-collected worms were subjected to a combinatorial approach comprising several standardized water formulations, three cultivation temperatures and several different sustenance foods to identify suitable husbandry conditions (Supplementary Fig. 1b). As a result, we now maintain a diverse planarian live collection (Fig. 1b,c and Supplementary Fig. 1c–e). Besides freshwater planarian species diversity, the collection also harbours distinct population isolates of several species (for example, S. polychroa isolates from across Europe, including an unpigmented isoline (Fig. 1b)) and some marine species that serendipitously proved amenable to laboratory culture (Fig. 1c). The Prolecithophora, as an outgroup to planarians (Fig. 1d), are not permanently hosted in the collection. Our collection represents most major planarian taxa and thus constitutes a coarse-grained sampling of extant planarian biodiversity.

We next took advantage of our species collection to systematically compare regeneration abilities under standardized laboratory conditions. We focused on head regeneration as arguably the most complex regeneration challenge and eye regeneration as a convenient morphological marker for head-regeneration success. Inspired by Šivickis's pioneering studies[23], we assayed the fractional success

of head regeneration at five transverse cuts equally spaced along the A–P axis (Fig. 2a; Methods). Regeneration defects were highly reproducible for a given species and, where applicable, mostly consistent with previous literature reports. We identified three broad groups of head-regeneration capacities (Fig. 2a,b). Group A, or 'robust regeneration', comprises species with ≥80% head-regeneration success at all cut positions. Plots of the head-regeneration frequency along the A–P axis, so-called 'head frequency curves'[24], are consequently flat and uniformly high (Fig. 2b), even though the rate of head regeneration may vary in a position-dependent manner (for example, in Smed[37]). The current planarian model species D. japonica and Smed (both the fissiparous and the egg-laying strain (Fig. 2a and Supplementary Fig. 2a)) are group A members, reflecting their robust and rapid regeneration. Group B, or 'restricted regeneration', comprises species with positional head-regeneration defects, that is <80% head-regeneration efficiency at one or more A–P positions (Fig. 2a,b). This group includes freshwater species with known head-regeneration deficiencies in the posterior body half (for example, Dendrocoelum lacteum)[23], several so far undescribed regeneration defects (for example, Cura pinguis) and also some marine species (for example, Cercyra hastata). The 'restricted regeneration' group also includes several cases of reduced regeneration efficiency in central body regions, for example, Dugesia sp. (Fig. 2a,b). Group C, or 'poor regeneration', comprises species incapable of head regeneration in any amputation fragment under our experimental conditions, with the marine planaria Bdeloura candida as a known example (Fig. 2a,b)[23].

Out of the total of 36 analysed planarian field isolates, four were poor regenerators (all different marine species), 14 displayed restricted regeneration and 18 were robust regenerators (Fig. 2c and Supplementary Table 1). Notably, our collection also harbours examples of each class in Šivickis's more fine-grained regeneration classification scheme[23], thus providing the intended broad overview of planarian regeneration and regeneration defects (Fig. 2b). Further, the collection also holds multiple examples of interesting species-specific regeneration peculiarities, such as a propensity for bipolar head or tail regenerates in Spathula sp. and Dendrocoelum lacteum or frequent aberrant axis duplications in mediolateral regenerates of one field isolate of Dugesia sicula (Fig. 2d). Interestingly, we also found the model species D. japonica to regenerate double-headed individuals with low frequency (Supplementary Fig. 2b), which is relevant regarding the interpretation of previously reported regeneration experiments in Earth's orbit[38,39]. Although not captured by our classification scheme, each of these species-specific regeneration peculiarities represents a future opportunity for probing the mechanistic underpinnings of planarian regeneration.

To put the regenerative abilities of planarians into a phylogenetic perspective, we further obtained and assayed live specimens of two Prolecithophora species (Fig. 2e) as representatives of the closest available Tricladidan sister taxon[40,41] (of the Fecampiidae, neither live specimens nor published information was available). Due to their small size (<1 mm length), head regeneration was only

**Fig. 2 | Quantitative analysis of head-regeneration abilities across the flatworm collection. a**, Cartoon of the serial head-regeneration assay (left) and representative outcomes in the indicated species. Dashed lines: amputation planes and typical regeneration outcomes (small images) and their relative frequency (number pairs; 'Dead' in case of no survivors) at the respective A–P position. Colours designate the head-regeneration classification scheme used throughout this study: group A (green): robust regeneration (efficient head regeneration at all A–P axis positions); group B (orange): restricted regeneration (position-dependent head-regeneration defects); group C (red): poor regeneration (no head regeneration at any A–P position). Missing images in the C. hastata panel correspond to pieces that did not regenerate a head and died before imaging (No head). **b**, Graphical representation of the data in the form of head frequency curves. The percentage of successful head regeneration

is plotted either as a fraction of the initial (black line) or surviving (red line) fragment numbers. Dashed line, percentage of fragment survival. The Roman numerals below designate the more fine-grained head-regeneration scheme by Šivickis[23]. **c**, Left, head frequency curves (percentage of surviving fragments) for all analysed species and colour-coded as in **a**. Right, number of species in each head-regeneration category. **d**, Unclassified regeneration defects and their relative frequencies in the indicated collection species, including bipolar double-heads (left), double-tails (centre) and mediolateral axis duplication (right). Amputation paradigms are cartooned to the left; the respective regeneration outcome in Smed is shown for reference. **e**, Phylogenetic overview of flatworm groups related to planarians[40] (left); amputation paradigm (centre); and representative images and quantifications of head-regeneration failures in the indicated Prolecitophora species (right). Scale bar, 1 mm unless otherwise noted.

assayed upon medial bisection. Consistent with a recently published analysis[42], we found the assayed Prolicetophora species incapable of de novo head regeneration, even though they can repair injuries to an existing head similar to *Macrostomum* or polyclads[43,44]. The rich and varied regeneration of planarians, therefore, contrasts with so far poor regeneration abilities of the sister groups, thus making the evolutionary history of regeneration in planarians particularly interesting[34].

## Evolutionary history of planarian head regeneration

Although multiple phylogenies of the Tricladida have been published to date, they are either based on morphology or cover few species or gene sequences[28]. To comprehensively reconstruct the phylogeny of our planarian species collection, we first de novo assembled transcriptomes via our published pipeline (Fig. 3a)[45,46]. Additionally, we incorporated publicly available or contributed data from eight species, yielding a total dataset of 49 transcriptomes (45 planarians and

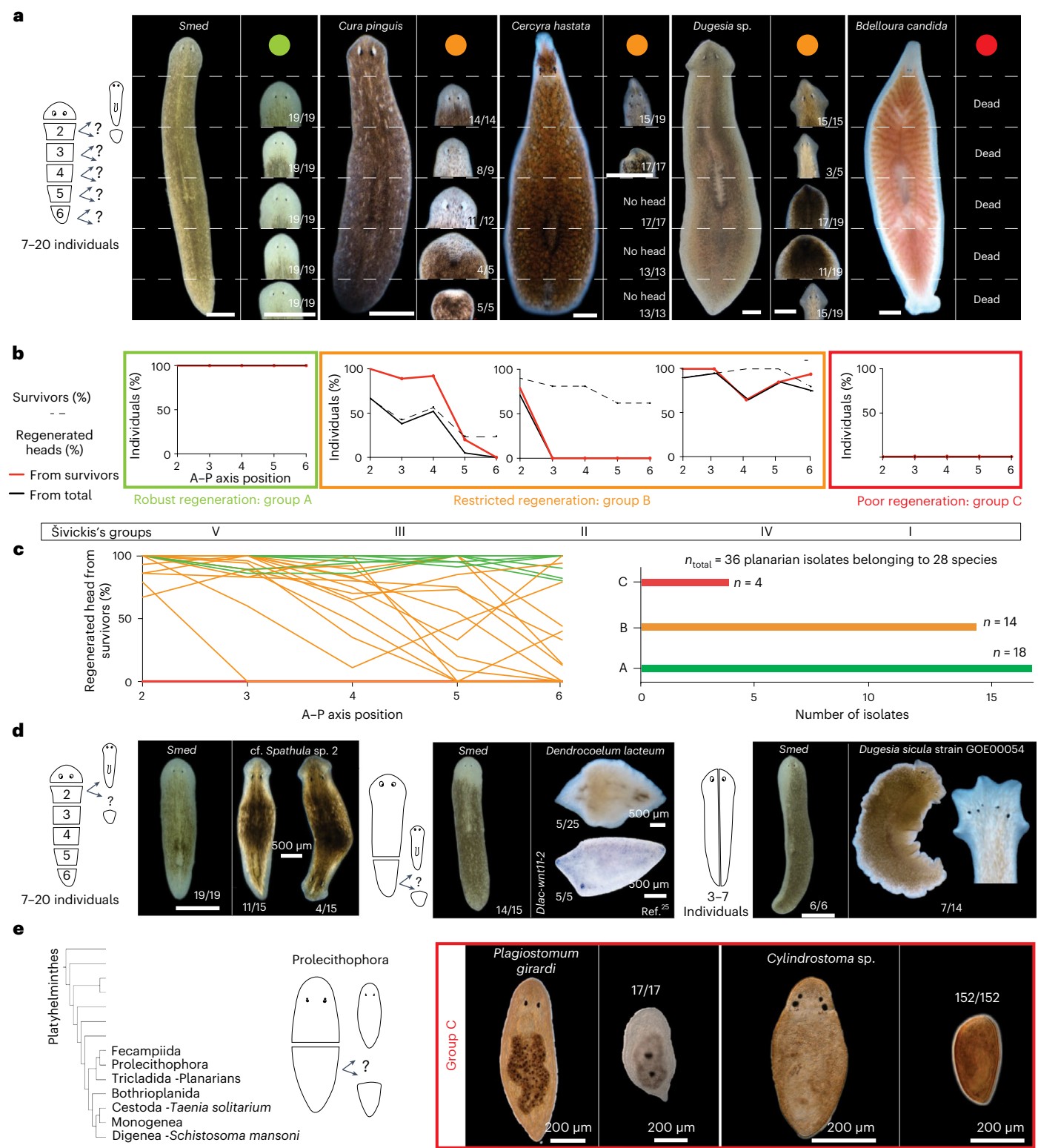

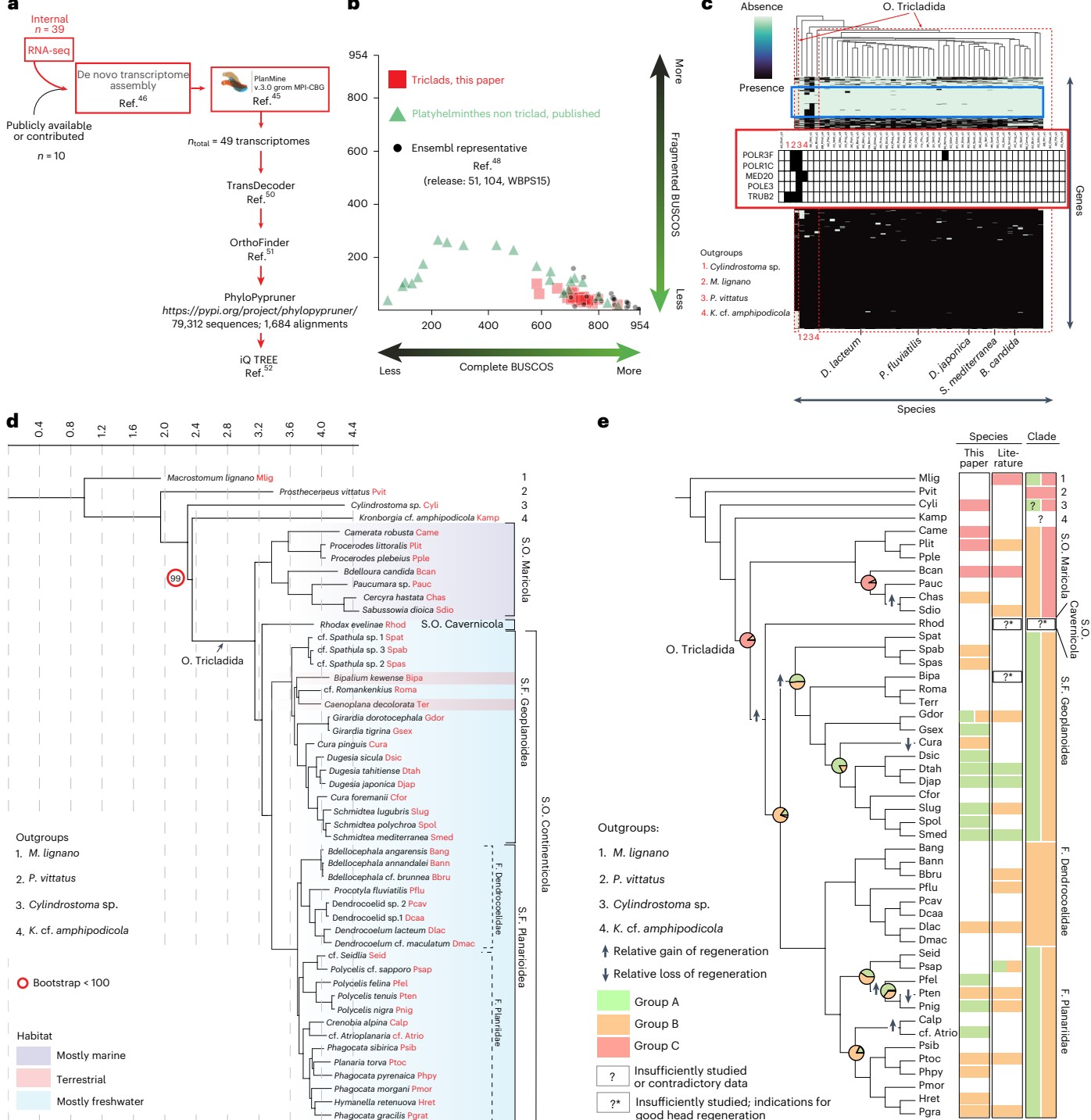

**Fig. 3 | Phylogenetic analysis of head-regeneration abilities in planaria.**
**a**, De novo high-quality transcriptome assembly pipeline and unbiased gene sequence selection workflow for multigene phylogeny reconstruction. RNA-seq, RNA sequencing. **b**, BUSCO quality comparison between the indicated transcriptome sources, plotting BUSCO gene sequence completeness (*x* axis) versus fragmentation (*y* axis). 'Perfect' BUSCO representation: bottom right corner. **c**, Presence (black)/absence (turquoise) analysis of individual BUSCO genes (*y* axis) in flatworm transcriptomes (*x* axis, in phylogenetic order) and representative outgroups. Names of representative species are indicated (see Supplementary Table 2 for detail). The blue frame designates BUSCO genes absent from >90% of the transcriptomes. The red inset shows a selection of likely planarian-specific BUSCO loss events. **d**, Maximum-likelihood tree on the basis of the transcriptomes with representatives of all the suborders of the

Tricladida. Numbers indicate outgroups and the preferred habitats of major planarian taxonomic groups are indicated by colour shading (see legend).
**e**, Phylogenetic map of planarian head-regeneration abilities and ASR. Pie charts summarize the most likely regeneration ability at selected internal nodes (for other nodes, see Supplementary Fig. 3f). Arrows indicate a parsimonious history of gains and losses of regeneration ability given the ASR. Nodes represent the proportion of character histories with the indicated state for regenerating capability (Supplementary Fig. 3f and Supplementary Tables 3, 4 and 5; Methods). The colour coding of the columns to the right indicates species- and clade-specific head-regeneration abilities quantified by this study or extracted from the literature (both used for clade annotation). See Supplementary Table 1 for the species abbreviations used in the figure. O, Order; S.O., Suborder; S.F., Superfamily; F., Family.

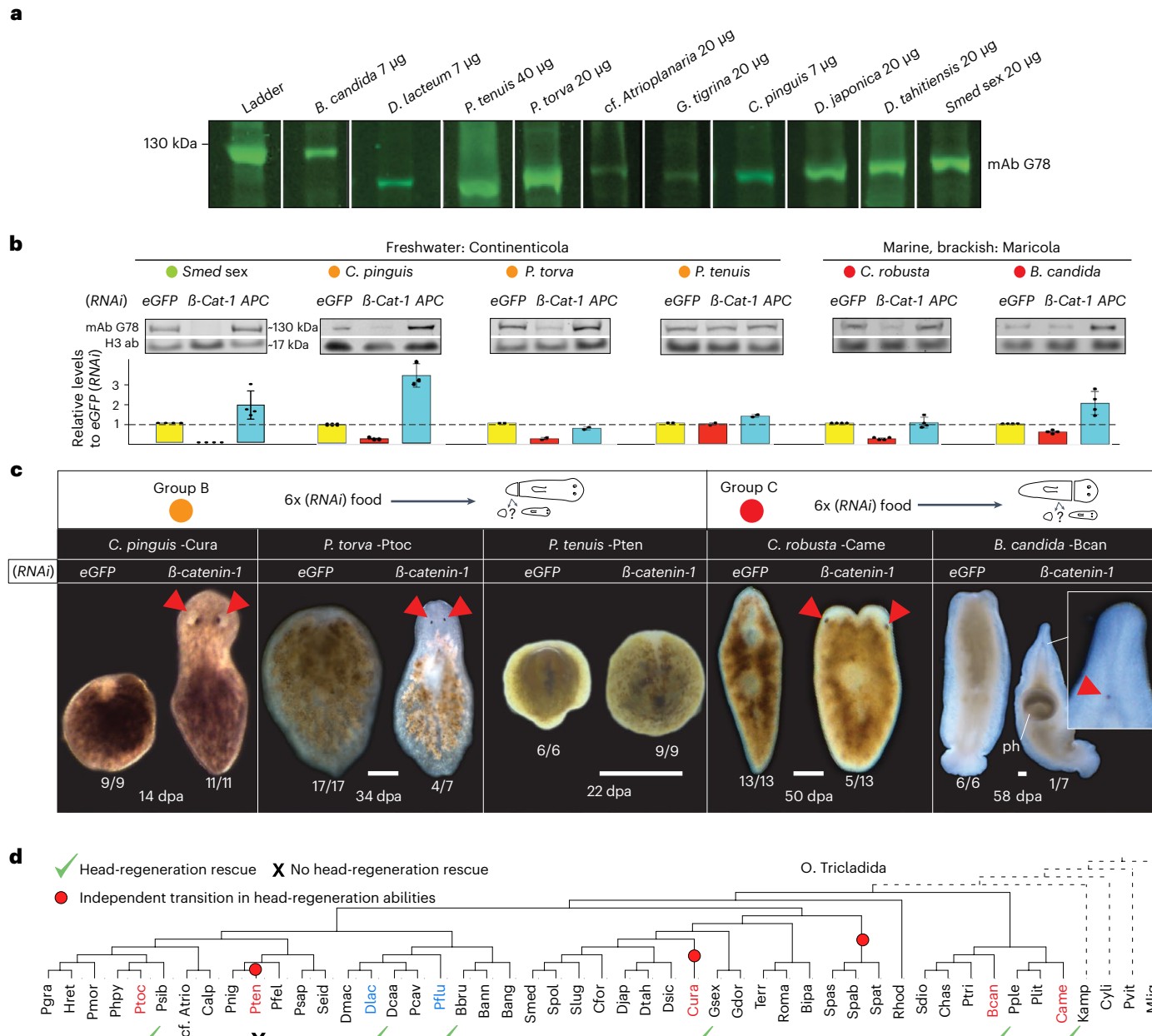

**Fig. 4 | Canonical Wnt pathway inhibition rescues head-regeneration defects across planarian phylogeny. a**, Fluorometric Western blot demonstration of anti-ß-CATENIN-1 antibody G78 crossreactivity with tail tip lysates of different planarian species. Species and amount of lysate/lane as indicated. **b**, RNAi-mediated Wnt pathway activity modulation in the indicated species from regeneration groups A, B or C (colour coding). Animal cohorts were treated with enhanced green fluorescent protein (*eGFP*) (*RNAi*) (control), *APC*(*RNAi*) (gain of Wnt signalling) or *ß-catenin-1*(*RNAi*) (loss of Wnt signalling; Methods). Top, representative quantitative Western blots for ß-CATENIN-1 and histone H3 as loading control. Bottom, bar graph representation of G78 signal intensity relative to the control in each species (*eGFP*(*RNAi*)). Data are presented as mean ± s.d. of

four technical replicates (dots). **c**, Head-regeneration rescue assay upon Wnt inhibition/*ß-catenin-1*(*RNAi*) in the indicated category B or C species. Live images at the indicated day postamputation (dpa) and RNAi conditions; number pairs represent the observed frequency of the shown phenotype; head or eye regeneration. Red triangles, regenerated eyes. Scale bar, 500 μm. ph, pharynx. **d**, Phylogenetic representation of documented head-regeneration rescued by *ß-catenin-1*(*RNAi*) (green check-mark) (phylogeny extracted from Fig. 3d). This study (red): *C. pinguis*, *P. torva*, *C. robusta* and *B. candida*. Previous studies (blue): *D. lacteum*[25] and *P. fluviatilis*[26]. *Phagocata kawakatsui*, for which head-regeneration rescue was also reported[27], was excluded due to the lack of a publicly available transcriptome. *ß-Cat-1*, *ß-catenin-1*.

four outgroup Platyhelminthes). BUSCO (benchmarking universal single-copy orthologues) analysis[47] broadly confirmed the high assembly quality of our transcriptomes in comparison with all other available flatworm transcriptomes in ENSEMBL[48] and selected outgroups. Our analysis revealed 109 BUSCOs missing from 90% of the analysed flatworm transcriptomes (Fig. 3c), yet 74 were present in the more basal flatworm representatives *Macrostomum lignano* and/or *Prosthoceraeus vittatus* (Fig. 3c, inset for examples; Supplementary Table 2) suggesting

subsequent gene losses in planarians and sister taxa consistent with the previously reported substantial gene loss in the *Smed* genome[49]. Overall, our analysis highlights the need for better integration of so far undersampled taxonomic groups into BUSCO and the high assembly quality of our flatworm transcriptomes.

To infer the lineage relationships between our collection species, we extracted broadly conserved single-copy orthologues from our transcriptomes[50–52] (Fig. 3a). Accordingly, the maximum-likelihood

representation of the phylogenetic tree shows very high support for all branches (Fig. 3d), which was further confirmed by a coalescent-based tree estimation strategy (Astral; Supplementary Fig. 3a). Our phylogeny includes so far poorly investigated planarian genera (for example, *Cura*, *Camerata* or *Hymanella*) and thus represents one of the most extensive taxon samplings to date. Although the inferred tree topology is broadly consistent with previously published trees of the major planarian clades[28], an interesting exception is the placement of the taxon Cavernicola as a sister group to the Continenticola (Fig. 3d) instead of the recently proposed sister group relationship to Maricola[53]. Although so far based on a single species representative, the high number of analysed genes and the high bootstrap value (100) support this new phylogenetic proposal. Further noteworthy insights include the relationship between terrestrial planarians and the freshwater genus cf. *Romankenkius*[54] or the split of the two *Cura* species, which therefore do not form a monophyletic group. Moreover, our deep sequence comparisons revealed generally deep splits between planarian lineages. Quantitative branch length comparisons confirmed an unusual degree of sequence divergence at least on par with that of the highly divergent nematodes (Supplementary Fig. 3b), which may reflect unusually rapid rates of genome evolution and/or the old age of planarian lineages. Besides its immediate use for taxonomic and phylogeographic studies, our comprehensive phylogeny provides a general basis for analysing trait evolution in planarians.

To reconstruct the evolutionary history of planarian head regeneration, we combined our phylogeny and quantitative head-regeneration analysis (and further literature mining[23,34,55–57]) for an ancestral state reconstruction (ASR) (Fig. 3e and Supplementary Fig. 3f). Within the context of head-regeneration capable species in two distant flatworm clades (Catenulida and Macrostomorpha; Supplementary Fig. 3g)[44] and the limited information on the regenerative abilities of more closely related sister groups, ASR best supports the following trait evolution scenario: poor regeneration (red) was ancestral, followed by the acquisition of restricted (regionally limited) regeneration (orange) in the Continenticola and some marine lineages and, finally, the acquisition of robust position-independent head regeneration (green) exclusively in the freshwater lineages. Transitions between poor regeneration and restricted regeneration are rare in our dataset (2) and only observed in the basal marine species (Maricola) (Fig. 3e and Supplementary Tables 3, 4 and 5). Transitions between 'restricted regeneration' and 'robust regeneration' appear limited to the Continenticola but probably occur frequently and in both directions (particularly in the Planariidae; Fig. 3e and Supplementary Table 4). Of note, our present taxon sampling cannot answer the question of whether de novo head regeneration is ancestral in the Platyhelminthes, which will ultimately require mechanistic comparisons of the head-regeneration process between Catenulids, Macrostomorpha and planarians. However, for planarians,

our data demonstrate an unexpectedly dynamic evolutionary history of head-regeneration capabilities, with at least five independent gains and three independent reductions of head-regeneration capability in our present species survey (Fig. 3e and Supplementary Table 3).

## Wnt inhibition rescues all head-regeneration defects

Independent evolutionary transitions in head-regeneration abilities imply independently evolved changes in the underlying molecular control network. So far, Wnt pathway misregulations have been linked to the head-regeneration defects of two dendrocoelids (*D. lacteum* and *Procotyla fluviatilis*) and the planariid *Phagocata kawakatsui*[25–27]. Our species collection provided an opportunity to systematically examine the contribution of Wnt signalling to planarian head-regeneration deficiencies. Planarian canonical Wnt signalling activity can be monitored on the basis of ß-CATENIN-1 amounts due to the segregation of the signalling and cell adhesion roles of *ß-CATENIN* between different gene homologues[58] and our previously characterized anti-*Smed*-ß-CATENIN-1 antibody[16] raised against the highly conserved Armadillo repeats detected a band of the expected size in multiple species (Fig. 4a and Supplementary Fig. 4c). To experimentally modulate Wnt pathway activity, we used established RNA-mediated genetic interference (RNAi) feeding protocols to target the respective species-specific *ß-catenin-1* (Supplementary Fig. 4a,b) and *Adenomatous Polyposis Coli* (*APC*; negative pathway regulator) homologues (Fig. 4b). *C. pinguis*, *Planaria torva*, *C. robusta* and *B. candida* displayed the expected decrease or increase in relative ß-CATENIN-1 levels in response to the RNAi treatments[16], confirming the principal utility of the RNAi approach for experimental alterations of Wnt pathway activity in those species. Interestingly, *P. tenuis* was largely refractory to RNAi by feeding (Fig. 4b and Supplementary Fig. 4d), thus providing a first indication that not all planarian species may be equally susceptible to systemic RNAi, as observed in nematodes[59,60]. Interestingly, *ß-catenin-1*(*RNAi*) rescued head regeneration or promoted the appearance of head-like structures in all examined species except the RNAi-deficient *P. tenuis* (see above), yet with varying efficiencies (for example, 11/11 in *Cura pinguis* and 1/7 in *B. candida*). Even the lower rescue efficiencies were relevant due to the complete absence of similar phenotypes in the corresponding control pieces and, therefore, probably reflect the varying knockdown efficiencies in the different species (Fig. 4c). Intriguingly, we even observed indications of head-like tissue formation in the two Maricola species, which are probably ancestral group C members (poor regeneration). When considered in the context of planarian phylogeny and published data (Fig. 3e), our results indicate that the inhibition of canonical Wnt signalling via *ß-catenin-1*(*RNAi*) can generally bypass head-regeneration defects in planarians, independent of the evolutionary history of the specific lineage (Fig. 4d). By extension, these results further imply that head-regeneration defects across planarian phylogeny may be

**Fig. 5 | Wnt signalling functions in the *Smed* reproductive system. a**, Co-occurrence of robust head regeneration (Group A, red) with fissiparous reproduction across planarian clades. **b**, Live images illustrating the sexual (egg-laying; left) and asexual (fissiparous; right) reproduction modes of *Smed* laboratory strains. **c**, Cartoon (top) and colorimetric whole-mount in situ hybridizations (indicated markers; middle) of *Smed* sexual strain reproductive system components; reproductive system ablation under *ophis*(*RNAi*) (bottom). **d**, ß-CATENIN-1 amounts/Wnt signalling activity in A–P sections (1, head; 6, tail) quantified via Western blotting with the G78 mAB. *Smed* strain and RNAi conditions as indicated. Error bars, s.d. of *n* = 4 biological replicates, each representing the mean of four technical replicates (blots). **e**, Wnt component expression in the *Smed* reproductive system by colorimetric or fluorescent whole-mount in situ hybridization. Number pairs, specimen fraction displaying the pattern shown; red arrowhead, testes lobules; black arrowhead, expression in non-reproductive tissues; H3P, Histone H3 Ser10 phosphorylation immunolabelling; DAPI, nuclei. **f**, Colorimetric whole-mount in situ hybridizations of the indicated reproductive system markers under the indicated

RNAi conditions; number pairs, specimen fraction displaying the pattern shown; dashed lines, approximate position of the sagittal sections in Fig. 5g. **g**, Relative yolk gland cross-sectional area quantifications in histological sections of the indicated RNAi conditions. Top, bar graph, *n* = 2 individuals (dots)/condition; error bars, s.d. of the mean. Bottom, representative Mallory-stained sagittal sections. White outline, yolk glands; red boxes, zoom views. y, yolk; vnc, ventral nerve cord. **h**, Coomassie-stained SDS–PAGE of lysates of the indicated sources. White box, major yolk protein. **i**, Fluorescent Western blot of the indicated lysates, probed with the indicated antibodies. **j**, EO95 specificity analysis in *Smed* sexual strain lysates under the indicated *ferritin*(*RNAi*). Asterisk, EO95 signal loss in *ferritin-C*(*RNAi*). **k**, Control (*eGFP*)-normalized quantification of FERRITIN-C and ß-CATENIN-1 amounts in lysates of the indicated RNAi condition via EO95 and G78 immunoblotting. Error bars, s.d. of the mean of four technical replicates (dots) in *n* = 1 biological replicate. **l**, Colorimetric whole-mount in situ hybridization of the four yolk *ferritins* in sexual *Smed* under the indicated RNAi treatments. Scale bar, 1 mm unless otherwise noted.

associated with a functional excess of Wnt pathway activity and that this pathway is therefore a putative hot spot in the evolution of planarian head-regeneration defects.

## Wnt signalling requirments in the *Smed* reproductive system

Regeneration-deficient planarian species are not laboratory mutants. Therefore, natural selection mechanisms must exist that drive the repeated emergence of regeneration defects. Previous authors have suggested that regeneration in planarians and other clades may be under selection as a necessary aspect of asexual reproduction by fission[44,61] and, therefore, might become dispensable in egg-laying

species. In this context, a positive pleiotropy of Wnt signalling in egg-dependent reproduction provides an intriguing hypothesis to explain the repeated emergence of Wnt-dependent regeneration defects in egg-laying species.

As a first test of this idea, we examined the predicted cosegregation of regenerative abilities with reproduction modes. As shown in Fig. 5b, poor or restricted regeneration is invariably associated with egg-laying, while fissiparous strains or species are almost invariably robust regenerators (Fig. 5b; the mild central body regeneration restriction of some dugesids constituting the only exception). Furthermore, poor or restricted regeneration species tend to occur in stable habitats

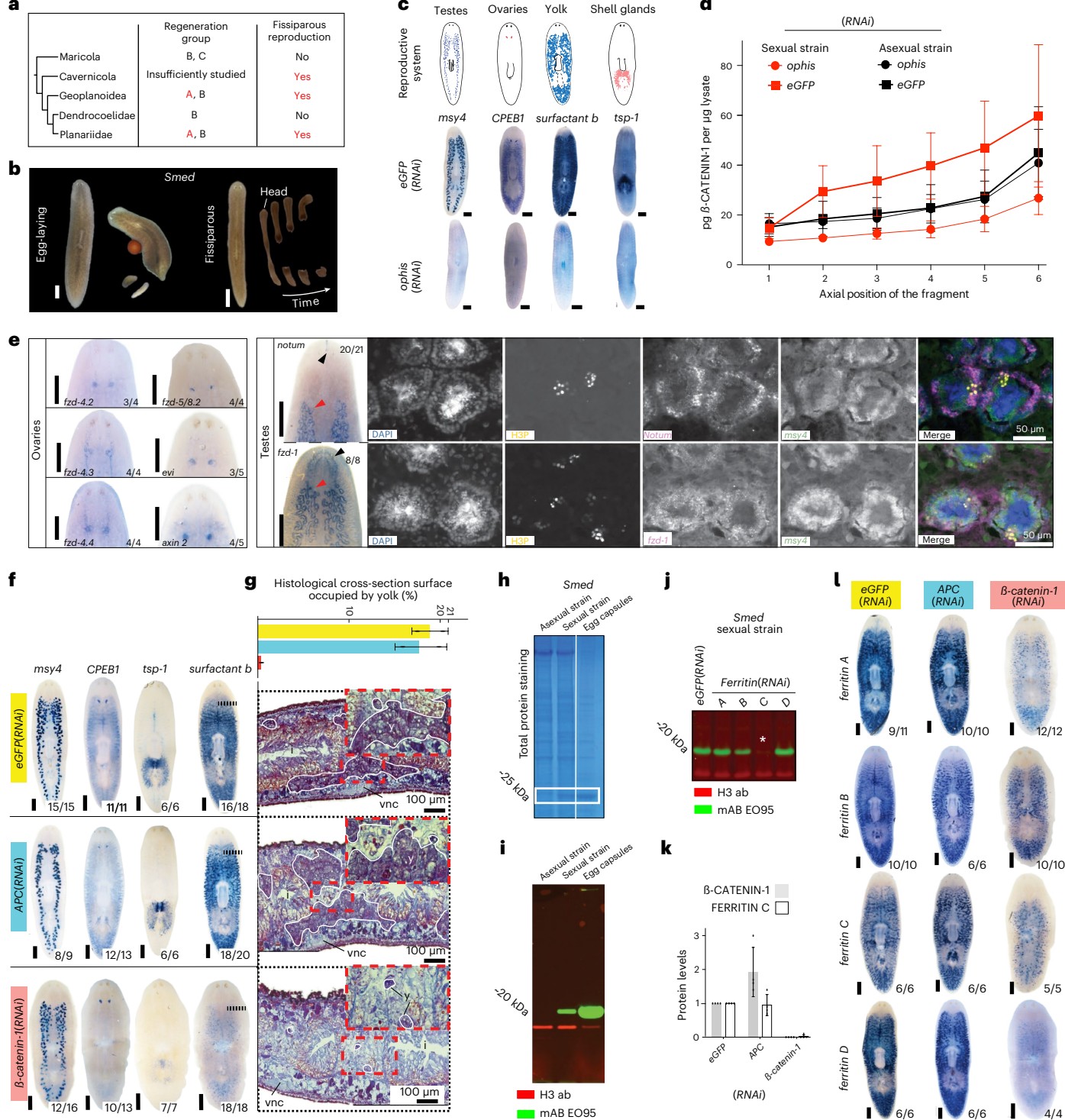

(for example, lakes or the sea), while robustly regenerating fissiparous strains or species tend to occur in less stable habitats (for example, fast-flowing streams or temporary water bodies (Supplementary Fig. 5a; refs. [62],[63])). Hence these observations are consistent with the selection of robust whole-body regeneration as part of fissiparous reproduction, which may become dispensable in habitats that favour egg-laying species.

To begin exploring the potential for pleiotropy of Wnt signalling in the planarian reproductive system, we turned to the sexual laboratory strain of *Smed*. Unlike the more commonly studied fissiparous (asexual) strain, adults of the sexual strain reproduce via depositing big egg capsules (Fig. 5b), and develop a hermaphroditic reproductive system including testes, ovaries and yolk glands[64],[65] (Fig. 5c) that can be collectively ablated through the knockdown of the orphan G-protein coupled receptor *ophis*[66]. As a first test of Wnt signalling activity in association with the reproductive system, we compared ß-CATENIN-1 levels between sexual and asexual *Smed* strains using our established quantitative Western blotting assay[16] (Fig. 5d). Interestingly, Wnt signalling levels were consistently higher in the sexual strain and *ophis*(*RNAi*) reduced ß-CATENIN-1 levels to asexual-like levels, which indicates substantial canonical Wnt signalling activity in association with the *Smed* reproductive system. This finding is consistent with previous findings in the *Smed* sister species *S. polychroa*[67] and with the expression of several Wnt receptors and other pathway components in multiple reproductive system-associated tissues (Fig. 5e and Supplementary Fig. 5b,c).

To probe the functions of Wnt signalling in the reproductive system, we activated (*APC*(*RNAi*)) or inhibited (*ß-catenin-1*(*RNAi*)) Wnt signalling in sexually mature *Smed* and subsequently assayed the expression patterns of reproductive system marker genes (Fig. 5f and Supplementary Fig. 5d; note that specimens were examined before overt body plan transformations)[16]. While testes and ovaries appeared overtly normal (but note indications of testes lobe alterations in *APC*(*RNAi*) (Supplementary Fig. 5d)), the expression of both yolk and shell gland markers was practically undetectable under *ß-catenin-1*(*RNAi*), yet denser in *APC*(*RNAi*) (Fig. 5f and Supplementary Fig. 5e). Moreover, quantifications of the relative cross-sectional area of the yolk glands in histological sections of mature control, *ß-catenin-1* and *APC*(*RNAi*) animals revealed the near-complete ablation of the yolk glands by *ß-catenin-1*(*RNAi*) (Fig. 5g). Given the general importance of yolk as a life history parameter, we first performed gel electrophoresis of egg capsule extracts (Fig. 5h) and subsequent mass spectrometry to identify FERRITINs as the major yolk protein in *Smed* as in *Dugesia ryukyuensis*[68] and raised a monoclonal antibody against gel-eluted FERRITINs as direct read-out of yolk amounts (Fig. 5g; Supplementary Fig. 5g). Consistently, Wnt inhibition via *ß-catenin-1*(*RNAi*) (Fig. 5k) resulted in the near-complete depletion of yolk protein and the expression of the corresponding genes (Fig. 5i-l); while *APC*(*RNAi*) induced seemingly denser yolk gland labelling (Fig. 5l). Overall, our results demonstrate important Wnt signalling functions in the *Smed* reproductive system, including a quantitative influence on yolk content.

## Model for the evolution of head regeneration in planarians

Our results are consistent with the following working model for the evolution of planarian head regeneration (Fig. 6a, centre): robust whole-body regeneration in planarians is necessary for fissiparous reproduction but dispensable for reproduction via egg-laying (centre). Each reproductive strategy has different costs and benefits in specific environments (Fig. 6a, red arrows). Fissiparous (asexual) reproduction avoids the cost of males and allows the establishment of a new population from a single individual[69],[70], which may be particularly advantageous for the rapid colonization of temporary habitats or fast-flowing mountain streams. Egg-laying and the commonly entailed meiotic recombination generate new allele combinations that may benefit long-term population survival in stable habitats, for example, large lakes or the sea. Selection for each of the two reproductive strategies entails opposite selective pressures on planarian Wnt pathway activity (top); whereas reproductive performance via egg-laying is positively influenced by high Wnt pathway activity (for example, yolk production in *Smed*), excess pathway activity interferes with whole-body regeneration and, thus, fissiparous reproduction. Hence the model envisages the emergence of planarian regeneration defects via the habitat-specific selection for extreme egg capsule size or production rate and concomitant elevation of Wnt signalling levels to a point where they begin to interfere with head regeneration. The many egg-laying and whole-body regeneration-competent species (for example, the sexual *Smed* strain) are envisaged to occupy intermediate levels of the trade-off regime, that is intermediate Wnt signalling levels and intermediate investment in sexual reproduction that do not yet interfere with regeneration.

The model makes several predictions: first, the Wnt-dependence of yolk production should be conserved across planarian phylogeny. Second, amongst egg-laying species, regeneration-restricted species should generally invest more in reproduction than robust regenerators. Third, Wnt signalling levels across planarian phylogeny should positively correlate with egg-laying versus fissiparous reproduction and peak in regeneration-deficient species. Our species collection and the tools generated during this study allowed a first experimental exploration of these predictions.

To ask whether the Wnt-dependence of yolk production is conserved across planarian phylogeny, we first established the broad interspecies yolk crossreactivity of the anti-FERRITIN-C mAb EO95 and signal specificity to sexually mature adults (Fig. 6b and Supplementary Fig. 6a,b). Interestingly, using the previously validated *ß-catenin-1* or *APC*(*RNAi*) knockdowns across our species panel, we failed to observe the expected *ß-catenin-1* dependence of the yolk content (Fig. 6b). To exclude the possibility of lineage-specific changes in the yolk ferritin gene complement, we additionally quantified the relative cross-sectional area of the yolk glands in histological sections of *C. pinguis* and *P. torva*. As shown in Fig. 6c,d, both species maintained their yolk glands under *ß-catenin-1*(*RNAi*), despite clear reductions of *ß-catenin-1* levels under the experimental conditions (Fig. 4b). While

**Fig. 6 | Model and model testing. a**, Model; see text for details. **b**, Top, quantitative Western blot analysis of yolk content (EO95 mAb; H3, loading control) in the indicated species and RNAi conditions. Bottom, bar graph representation of RNAi-control-normalized EO95 signal. Error bars, s.d. of two to four technical replicates (dots) of $n = 1$ biological replicate. **c**, Yolk gland cross-sectional area quantifications in sagittal sections of the indicated species and RNAi treatments. White oulines, yolk glands; red frames, zooms. **d**, Bar graph representation of **c**. Error bars, s.d. of the mean of $n = 4$ individuals (dots), each representing the mean of five technical replicates. **e**, Correlation between yolk gland cross-sectional area and regenerative abilities in the indicated egg-laying species. Species ordering by relative yolk content quantified as in **c,d**; colours: head-regeneration abilities (green, robust/group A; orange, restricted/group B; red, poor/group C). Error bars, s.d. of the mean of $n = 4$ individuals (dots), each representing the mean of five technical replicates. **f**, Statistical analysis of the data in **e**, replotted as log$_{10}$ of mean yolk content/individual in group A versus B and C species. Red line, distribution mean. Significance assessment via linear mixed model; the indicated $P$ value implies statistical significance. **g**, Calibration of the G78 mAb for interspecies comparisons. Western blot of recombinant His-tagged ß-CATENIN-1 fragments of the indicated species and probed with anti-penta-His Ab (top) or G78 mAb (bottom); ratio, species-specific correction factor. Molecular weight marker as indicated. **h**, Correlation between Wnt pathway activity, regenerative abilities (colour coding as in **e**) and reproduction mode (bottom) across the indicated species. Species ordering by ß-CATENIN-1 concentrations in tail lysates, as per corrected G78 mAB signal. Error bars, s.d. of the mean of $n = 3$ biological replicates (dots), each representing the mean of four technical replicates. **i**, Statistical analysis of the data in **h**, replotted as log$_{10}$ of the mean ß-CATENIN-1 tail tip concentration in fissiparous versus egg-laying strains/ species (left) or robust versus restricted or poor regeneration groups (A versus B and C). Red line, distribution mean. Significance assessment as in **f**; the indicated $P$ values imply lack of significance.

the Wnt signalling dependence of yolk production is therefore unlikely to be deeply conserved across planarian phylogeny, the possibility remains that other positive pleiotropies of Wnt signalling in the reproductive system (for example, shell gland specification or testes-specific functions) mediate the predicted positive association between Wnt pathway activity and the reproductive system in specific lineages.

To explore the predicted correlation of relative investment in sexual reproduction with regeneration defects, we quantified the relative cross-sectional area of yolk glands as a proxy. In the nine collection

species that consistently produce egg capsules under our laboratory conditions, the yolk glands of restricted or poor regeneration species (groups B/C) indeed tended to occupy a larger fraction of the cross-sectional area as in regeneration-competent species (group A; Fig. 6e). This tendency was statistically significant despite substantial interanimal variations in the yolk content in some species (for example, *P. tenuis*, *P. torva* or *Camerata robusta*) (Fig. 6f), which may reflect non-synchronous reproduction cycles under our laboratory culture conditions. These data are further consistent with

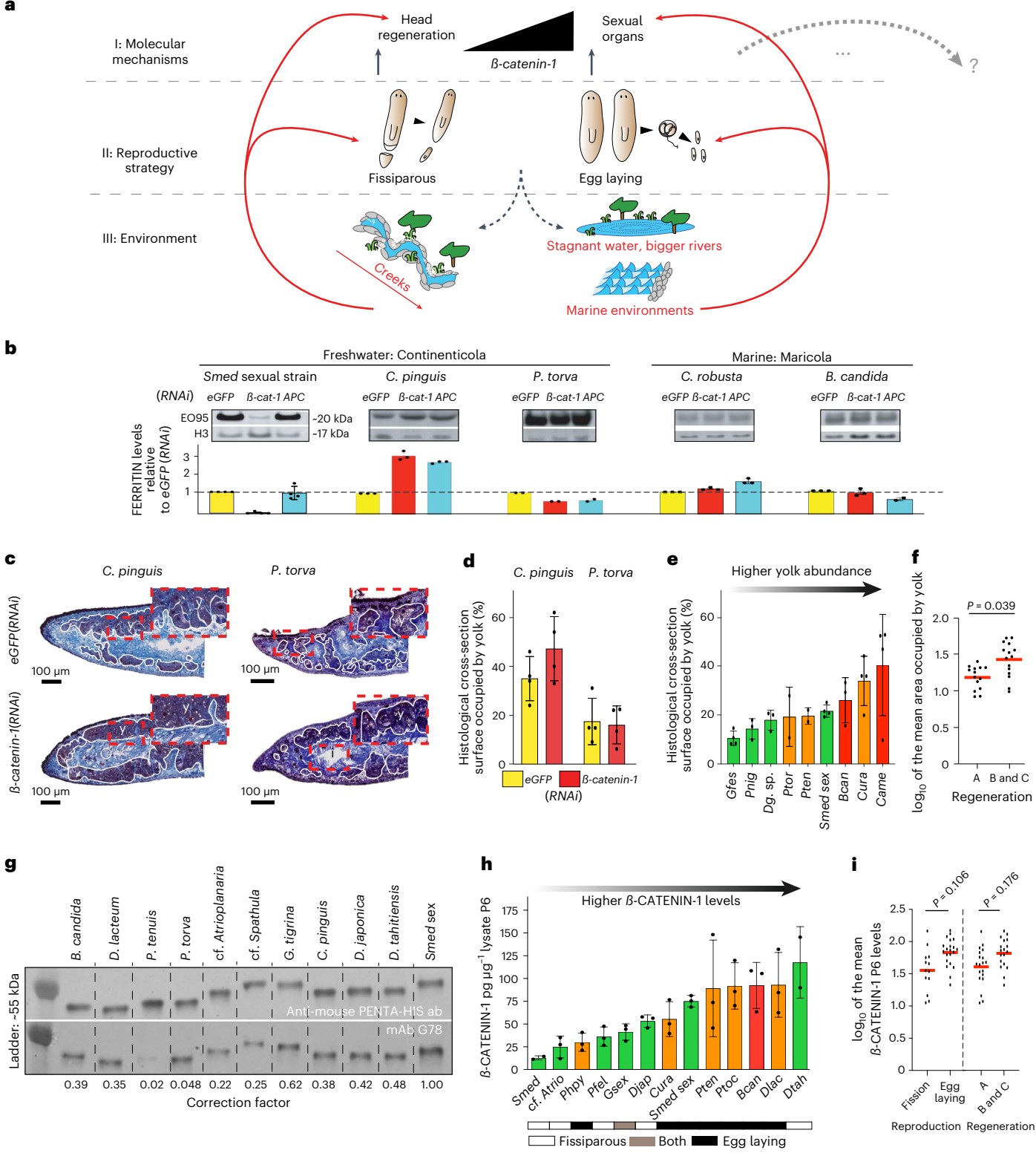

published reports of extraordinary investments in reproduction in the restricted regeneration species *D. lacteum* or *P. torva*[71], including the continuation of egg production even under starvation and extraordinarily large egg capsules in the case of *D. lacteum*. Hence these data are broadly consistent with the postulated higher investments in egg-laying reproduction in regeneration-deficient species compared to regeneration-competent ones.

Finally, to probe the degree of correlation between Wnt signalling levels and reproductive strategy or regeneration deficiencies, we calibrated our G78 pan-planarian anti-*ß*-CATENIN antibody as a proxy for interspecies comparisons of Wnt pathway activity. Using known amounts of in vitro translated recombinant Armadillo repeat fragments of the different species as quantitative Western blotting standards, we were able to measure and compare absolute ß-CATENIN-1 amounts in the lysates of different species (Fig. 6g and Supplementary Fig. 6c). We assayed a total of 13 collection species that represented a broad sampling of regenerative abilities, reproductive strategies and phylogenetic position (Fig. 6h). The ß-CATENIN-1 amounts in the tail tip samples of the different species varied by more than tenfold (Fig. 6h), suggesting considerable variations in signalling amplitudes between species. Interestingly, restricted or poor regeneration species were enriched amongst the species with the highest ß-CATENIN-1 levels. The statistical analysis of tail tip ß-CATENIN-1 levels relative to reproductive strategy (fissiparous versus egg-laying) or regenerative abilities (robust versus restricted and poor) similarly indicated a trend towards higher ß-CATENIN-1 levels in egg-laying and regeneration impaired species but such tendency was not statistically significant (Fig. 6i). Although the ß-CATENIN-1 quantifications in the present species panel failed to support the association between Wnt signalling levels, regeneration and reproduction, our model nevertheless remains a useful working hypothesis for guiding the future analysis of broader species samplings or cell biological analyses of Wnt signalling mechanisms in individual species.

## Discussion

Overall, our study cannot answer the question of why some planarians regenerate while others cannot. In fact, a single or simple answer may not exist in the face of the deep splits between planarian lineages and the associated divergence of molecular mechanisms that we discovered (for example, RNAi-susceptibility or the ß-CATENIN-1-dependence of yolk production). Nevertheless, our model of functional pleiotropies between Wnt signalling, regeneration and reproduction and selection of regeneration as a correlate of fissiparous reproduction provides a useful working model for the pursuit of a holistic understanding of planarian regeneration. Observations consistent with the model include the probable frequent transitions between robust and restricted regeneration in planarian phylogeny (Fig. 3), the experimental 'rescue' of independently evolved head-regeneration defects via the experimental inhibition of Wnt signalling (Fig. 4), multiple functional requirements of Wnt signalling in the reproductive system of the model species (Fig. 5) and the indications of correlations between Wnt signalling levels, regeneration and reproduction across the widely diverged planarian species (Fig. 6). On the mechanistic level, the assumed pivotal importance of Wnt signalling is plausible on the basis of the established pathway functions as both a molecular switch during head/tail regeneration and as provider of positional identity to the adult pluripotent stem cells (neoblasts) in non-regenerating intact animals[16–18]. Neoblasts generate and regenerate all planarian cell types[13], including all constituents of the reproductive system[72]. Hence an elevation of somatic Wnt signalling might upregulate yolk or testes formation while simultaneously interfering with head regeneration due to its dependence on Wnt inhibition. In general, our pleiotropy model assumes that regeneration and reproduction are under the influence of a common Wnt signalling source. Although our RNAi experiments are consistent with this premise, the lack of organ-selectivity of the current protocol leaves open the possibility of organ-autonomous Wnt signalling in the reproductive system and, therefore, functional and evolutionary uncoupling between reproductive and regenerative pathway regulation. Similarly, the model assumes a functional excess of steady-state Wnt signalling as a common cause of regeneration defects. Although again plausible on the basis of the *ß-catenin-1*(*RNAi*)-induced rescue of head-regeneration defects across planarian phylogeny, an alternative interpretation is that Wnt inhibition constitutes a deep developmental constraint in the neoblast-mediated formation of the planarian head. Accordingly, Wnt inhibition might be able to override all upstream physiological control mechanisms, inclusive of potentially Wnt-independent causes of head-regeneration defects[73–75] and thus bypass rather than rescue head-regeneration defects.

Above all, our working model stresses the need for a more fine-grained understanding of the cellular sources and targets of Wnt signalling, both during head regeneration and in the formation and maintenance of the reproductive system. In addition, a similarly fine-grained understanding of the mechanistic causes of head-regeneration failures in regeneration-deficient species will be necessary. The deep splits between planarian lineages and the lack of conservation of Wnt-dependence of yolk production beyond *S. mediterranea* indicate that results obtained in one planarian species may not necessarily apply elsewhere in the taxon, with the rapid diversification of molecular mechanisms in *Caenorhabditis* species as a case in point[76]. A further core premise of our working model is that Wnt signalling activity is the target of natural selection.

Clearly, this cannot be tested on laboratory populations and will require studies of natural populations and ecological niche factors (for example, predation). While illustrating the general challenge of linking proximate to ultimate mechanisms in evolution, these considerations highlight the experimental opportunities that planarians and our species collection offer to regeneration research.

## Methods

### Field collections

The Max Planck Institute for Multidisciplinary Sciences (MPI-NAT) planarian collection was assembled via dedicated field expeditions, mostly before 2015. Planarians were collected from the underside of stones, aquatic plants or other submerged objects with the help of a brush. Trapping was also done, using submerged plastic containers with an appropriately perforated lid and baited with liver (Supplementary Fig. 1a). The collected planarians were transferred to 50 ml Falcon tubes, maintaining low animal densities to avoid worm lysis. Tubes were kept cold and daily water changes with water from the collection site were carried out throughout the duration of the field campaign. The GPS coordinates and habitat features of collection sites were recorded in a dedicated database.

Prolecithophoran specimens were sampled from brown algae in the Adriatic sea in a harbour on the island of Krk, Croatia. The algal samples were first incubated for at least 15 min in a 1:1 7.14% $MgCl_2 \times 6H_2O$ and seawater and the solution was then filtered through a 63 µm pore-sized mesh. All animals retained in the mesh were washed with seawater into a Petri dish and prolecithophorans were collected with Pasteur pipettes under a stereo microscope. Until use for experiments, prolecithophorans were maintained according to Grosbusch et al.[77].

Collection species and specimens can be made available to members of the community upon request to the corresponding authors, subject to availability.

### Animal husbandry

To reduce the chance of inadvertent pathogen introductions, field-collected planarians were initially treated for 1–3 days with a cocktail containing the following reagents diluted in the appropriate species-specific culture media (final concentrations): rifampicin

at 100 µg ml⁻¹ (stock diluted in dimethylsulfoxide), erythromycin at 4 µg ml⁻¹, gentamycin at 50 µg ml⁻¹, vancomycin at 10 µg ml⁻¹, ciprofloxacin at 4 µg ml⁻¹, colistin at 20 µg ml⁻¹, penicillin/streptomycin/amphotericin B (Millipore 516104) 1:100 from stock and Spirohexol Plus 250 (JBL;1007100) (12 µl from stock per 10 ml of final solution). To establish long-term cultures of field-collected animals, one of four water formulations (see below) and three culture temperatures (10, 14 or 20 °C) were chosen as starting point according to the field notes. Montjuïc planarian water, the widely used salt solution for *Smed* laboratory cultures[36], was used for most planarian species. *Polycelis felina* water (PofW) is the 1× combination of two commercial aquaria salts, DRAK-Aquaristik GH+ (DRAK-Aquaristik, W0060100) and KH+ (DRAK-Aquaristik, W0070100). The 20× stock solutions of both products were mixed and diluted to the final 1× concentration of each salt with distilled water. Afterwards, 0.25 ml l⁻¹ of Tetra ToruMin (Tetra) was added to mimic organic solutes. Marine water was prepared by diluting Classic Meersalz (Tropic Marin, 10134) in MilliQ water to a final salt concentration of 32 g l⁻¹ with the help of a refractometer. Cave planarian water (CPW) is PW diluted to a final conductivity of 200 µS cm⁻¹ with MilliQ water and pH adjusted to 8.0 with 0.1 M NaOH. For species cultured at 10 or 14 °C, the water formulations were always precooled to minimize temperature shocks. To establish suitable food sources for wild-collected planarian species, small-scale feeding trials with organic calf liver paste, frozen and irradiated (60 Gy) mealworms (*Tenebrio molitor*) or rinsed *Drosophila* larvae were carried out and the food source eliciting the strongest feeding response was chosen as sustenance food. For *Smed*, the culture media was supplemented with gentamycin sulfate at 50 µg ml⁻¹. Episodic gentamycin supplementation was also used for other species in case of indications of poor culture health. Collection species were maintained in plastic dishes and fed at intervals with the sustenance food of choice for the species. Feeding was generally coupled with cleaning/rinsing with temperature-equilibrated water formulations. Feeding/cleaning intervals were adjusted depending on culture temperatures and feeding schedules.

### Live imaging
Flatworms were imaged with a Nikon AZ100M microscope equipped with a Digital Sight DS-Fi1 camera or a ZEISS Stereo Microscope Stemi 508 equipped with a Digital ZEISS Axiocam 208 colour camera. Only animals from healthy laboratory populations were chosen for the experiments. Prolecithophorans were imaged either with a Leica DM 5000 B microscope equipped with a Leica DFC 490 camera or on a Leitz Diaplan light microscope equipped with a DFK 33UX264 camera.

### Quantitative analysis of head regeneration
To quantify species-specific head-regeneration abilities, 7–20 specimens per strain were cut into six even pieces along the anteroposterior body axis. For species ~5 mm in length, animals were cut into four pieces (*Procedores littoralis*, *Procerodes plebeius* and *Camerata robusta*). Cuts were performed with the help of a microsurgical knife under a stereoscope. Animals were cold-immobilized on a wet sheet of filter paper using custom-built Pelletier cold blocks. *Dugesia sicula*, a temperature-sensitive species, was cut at room temperature. The resulting amputation fragments were grouped according to body axis position and maintained in Petri dishes or small plastic boxes in the accustomed maintenance medium of the species. Water changes were carried out daily for the first 3 days postamputation and every 3–4 days for the remainder of the experiment. Eye regeneration as a morphological marker of head regeneration was scored at 3–4 day intervals for a maximum of 8 weeks to account for species-specific variations in regeneration rates. Head regeneration at each position was quantified as the fraction of fragments (either initial number or surviving pieces) that successfully regenerated eyes. For Prolecithophora, the experimental setup for midbody amputations and monitoring of regenerates is described in detail in ref. 42.

### Transcriptome sequencing and assembly
RNA extraction and quality control were performed following an established protocol[16] and processed for 100 or 150 base pair paired-end Illumina sequencing. Double-indexing was used to minimize the cross-contamination of transcriptomes. Transcriptome assembly was carried out with our established pipeline[46]. Assembly completeness was assessed using BUSCO[47] (v.5.2.2, metazoa odb10—parameters: -protein). TransDecoder (v.5.5.0, parameters: --single_best_orf) was used for the in silico translation of the transcriptomes as input for BUSCO.

### Orthology inference and phylogenetic tree inference
An initial set of homologous groups were identified by applying OrthoFinder[51] (v.2.5.4, parameters: -M msa -I 1.5) to all proteomes (in silico-translated transcriptome assemblies). The resulting homologous groups were subsequently aligned using MAFFT[78] (v.7.487, parameters: --localpair --maxiterate 1000) and for each alignment, a gene tree was constructed using FastTree[79] (v.2.1.10). The alignments and their corresponding phylogenetic trees were provided as input to the tree-based orthology inference programme PhyloPyPruner (Sandberg et al. (manuscript in preparation), v.1.2.3, parameters: --min-len 150 –trim-lb 4 –min-support 80 –prune MI –min-taxa 65 –min-otu-occupancy 0.0 –min-gene-occupancy 0.0; github.com/fethalen/phylopypruner). The optimal parameters for PhyloPyPruner were chosen by comparing the outcome when adjusting for minimum sequence length, long branch trimming factor, minimum support value, minimum number of taxa, minimum operational taxonomic unit occupancy, tree pruning method and minimum gene occupancy. The optimization script, including the tested parameter values, can be found in the Supplementary Information. Phylogenetic trees were constructed using IQ-TREE[80] (v.2.1.2, parameters: -m MFP -bb 1000 -bnni) or via ASTRAL[81] (v.5.7.1), using standard parameter settings (Supplementary Fig. 3a). The phylogeny combining triclads, mammals and nematodes was built following the same approach as for the planarian phylogeny.

### Ancestral state reconstruction
The likely evolutionary history of regenerative abilities in planarians and the macroevolutionary transition rates between regeneration states was inferred using ASR. The analysis was performed on the basis of the experimental classification of species-specific regeneration abilities (26 cases) and the literature (18 cases) (Fig. 3e). Two species datasets were created to control for the population-dependent variation in head-regeneration ability in *Girardia dorotocephala* and *Polycelis sapporo*. Those two species are categorized in group A in dataset A and group B in dataset B (Supplementary Fig. 3c). The analysis was then performed on both datasets. Figure 3e shows the analysis for dataset A. ASR of regeneration ability was performed on the maximum-likelihood phylogeny, which was transformed to be ultrametric with a root depth of one using the penalized likelihood method[82] implemented in the software TreePL[83]. First, it was determined if the gain or the loss of regeneration ability occurred at different rates. For this, the appropriate transition matrix for ASR was determined by fitting MK-models with equal transition rates, with symmetric transition rates (SYM) and with all transition rates different. The best-fitting model was chosen using the corrected Akaike information criterion. SYM was the preferred model for both datasets. Second, stochastic character mapping[84] implemented in the R package phytools[85] was used to infer the most likely ancestral states at internal nodes of the phylogeny under the SYM model. The maximum-likelihood implementation of the method was used, which sample histories from the most likely transition matrix. After a burn-in of 10,000 iterations, 100,000 iterations of the stochastic sampling were performed, retaining every tenth character history resulting in 10,000 sampled histories. At each internal node, the proportion of histories with a given state reflects the likelihood that the ancestor at that node had this regeneration ability. Finally, the

total number of transitions across the phylogeny was summarized as the arithmetic mean of the number of changes in all 10,000 simulations. For an overview of the workflow, see Supplementary Fig. 3c–e.

### Identification of *ß-catenin-1* and *APC* in multiple planarian species

Homologues of *Smed ß-catenin-1* and *APC* in other species were identified using reciprocal BLAST against the respective transcriptome assemblies. For the assembly of the *ß-catenin-1* phylogeny, transcript sequences were translated using TransDecoder (v.5.5.0), (https://github.com/TransDecoder) and the single best open reading frame per sequence was selected. Translated sequences were aligned using MAFFT (v.7.490)[78], using --maxiterate 1000 --localpair as parameters. Next, trimAl (v.1.4.1)[86] was used with -automated1 and the tree was built using IQ-TREE (v.1.6.12)[80] with the parameters -m MFP -bb 1000 -bnni.

### Cloning and RNA-mediated gene silencing

For gene cloning, complementary DNA was synthesized using the SuperScript III Reverse Transcriptase (LifeTechnologies, 18080093) according to the manufacturer's recommendations, followed by an *Escherichia coli* RNase H step. DNA templates were amplified from cDNA using either published primers or primers designed using our transcriptomes (Supplementary Table 6) and cloned into the pPR-T4P vector by ligation-independent cloning[87] or the paff8cT4P vector (for recombinant protein production)[88]. For *Smed*, RNAi of specific target genes was carried out by feeding liver paste mixed with in vitro synthesized double-stranded RNA[89]. For RNAi experiments in species other than *Smed*, *Artemia* paste obtained via sonication of *Artemia* larvae was added to the liver paste (10% of the final RNAi food volume) to improve feeding efficiency. RNAi feedings were performed every third day, with a final dsRNA concentration of 1 or 2 µg µl$^{-1}$ of food. Animals received three feedings unless stated otherwise.

### Gene expression analyses

Riboprobe production, animal fixation, colorimetric in situ hybridization, FISH and mounting were performed largely as described[90] but incorporating several optimizations for large planarians (>1 cm)[91]. Representative specimens from colorimetric whole-mount in situ hybridization were imaged with a Nikon AZ100M microscope equipped with a Digital Sight DS-Fi1 camera. For the documentation of FISH, we used a Zeiss Axio Observer.Z1 with a confocal Zeiss LSM 700 scan head equipped with a ×20 or ×25 objective. Brightness/contrast and colour balance were adjusted using Adobe Photoshop Creative Cloud 2018 and always applied to the whole image. Figure panels were assembled using Adobe Illustrator Creative Cloud 2018 and Affinity Designer. For planarian live image montages (for example, Fig. 1), the animals were cropped out of the original image frame and pasted onto a uniform black background.

### Identification of *Smed* yolk proteins

Two-day old egg capsules were homogenized in 200 µl of cold RIPA buffer (150 mM NaCl, 50 mM Tris-HCl pH 8.1, 0.1% SDS (w/v), 1% sodium deoxycholate, 1% NP-40) with a pestle and pellet mixer and incubated for 20 min on ice. Laemmli buffer (6× Laemmli buffer; 12% SDS; 0.06% bromophenol blue; 50% glycerol; 600 mM dithiothreitol; 60 mM Tris-HCl pH 6.8) was added to a final concentration of 1× and incubated for 10 min at 95 °C. Six volumes of 1× Laemmli buffer were added and 10 µl of the sample was used for SDS–polyacrylamide gel electrophoresis (SDS–PAGE; Supplementary Fig. 5f). The bands in the gel were cut, homogenized and analysed via mass spectrometry by the MPI-CBG mass spectrometry facility.

### In vitro transcription-translation and antibody production

The Expressway Maxi Cell-Free *E. coli* Expression System was used according to the manufacturer's recommendations for the in vitro production of species-specific ß-CATENIN-1 standards. *ß-catenin-1* fragments of different species cloned into the paff8cT4P vector were used as a template. The concentration of the resulting His-tagged ß-CATENIN-1 fragments was quantified via quantitative Western blotting with a Penta-His antibody (Qiagen) and a recombinant His-tagged protein as external standard.

The FERRITIN antibody EO95 was raised against gel-eluted protein at the MPI-CBG Antibody Facility[16]. In short, 2-day-old egg capsules laid at 20 °C were homogenized in 700 µl of cold RIPA buffer (150 mM NaCl, 50 mM Tris-HCl pH 8.1, 0.1% SDS (w/v), 1% sodium deoxycholate, 1% NP-40) and then incubated for 10 min at 95 °C. Proteins were separated by SDS–PAGE and stained with KCl. The band containing FERRITIN was cut out and eluted by electro-elution into migration buffer (pH 8.5) containing 25 mM Tris, 0.19 M glycine, 0.1% SDS. The procedure was repeated until a protein concentration of 1.6 mg ml$^{-1}$ was reached. A total of 30 µg of protein was injected into a BALB/c mouse for immunization and 15 µg for boosting. Test bleeds, as well as final antibodies, were tested via Western blot using protein lysate from asexual and sexual *Smed* as well as egg capsule lysate. (All antibodies are available upon request to the corresponding authors via the MPI-CBG antibody facility).

### Quantitative Western blotting and analysis

Quantitative Western blotting was performed as described in ref. [16] with minor modifications. Animals were fixed in zinc fixative (100 mM ZnCl$_2$ in 100% EtOH) for 30 min at 4 °C and then stored at −80 °C. To prepare the complete lysis buffer, to 1 ml of freshly thawed 9 M urea lysis buffer (9 M urea, 100 mM NaH$_2$PO$_4$, 10 mM Tris, 2% SDS, 130 mM dithiothreitol, 1 mM MgCl$_2$, pH 8.0) we added phosphatase inhibitor (25 µl of 40× PhosSTOP), benzonase (10 µl of 250 unit ml$^{-1}$ solution, final concentration 1%) and protease inhibitor cocktail (20 µl of 100× HaltTM Protease Inhibitor Cocktail). Samples were run out in technical quadruplicates on NuPAGE Novex 4–12% Bis-Tris protein gels in 1× MES-SDS running buffer, transferred onto nitrocellulose membranes in 1× NuPage Transfer Buffer (25 mM Bicine, 25 mM Bis-Tris (free base), 1 mM EDTA, 0.05 mM chlorobutanol, 1 mM NaHSO$_3$, 0.01% SDS, 20% methanol, pH 7.2), blocked in 5% soya protein powder solutions in 1× PBS and incubated with primary antibody (anti-*Smed*-ß-CATENIN-1 G78 mouse monoclonal at 1 µg ml$^{-1}$; anti-*Smed*-FERRITIN EO95 mouse monoclonal at 0.1 µg ml$^{-1}$; anti-rabbit Histone H3 (H3) (Abcam, ab1791) at 10 ng ml$^{-1}$ in 5% soya protein powder in 1× PBS with 0.1% Tween20. Membranes were washed with washing buffer (1× PBS with 0.1% Tween20) before incubation with infrared fluorescent secondary antibodies (anti-Mouse 770CW, LI-COR and anti-Rabbit IRDye 680LT, LI-COR) diluted at 1:20,000 in blocking solution. Membranes were washed with washing buffer, followed by a final wash step in 1× PBS without Tween20. Stained membranes were dried and imaged on an LI-COR Odyssey imager.

All Western blot image quantifications were conducted using ImageStudioLite software (LI-COR). Rectangles were drawn around the protein band of interest (ß-CATENIN-1 or FERRITIN) and H3 (loading control) and then the background was subtracted from the total signal using the 'Median' method. The signal of each ß-CATENIN-1 band was normalized to the H3 band of the same lane as loading control. To obtain absolute ß-CATENIN-1 concentrations (pg µg$^{-1}$ of total protein lysate) from protein samples, 200, 800 and 1,500 pg of the respective recombinant ß-CATENIN-1 standard were run out on the same gels as the experimental samples. A three-point regression analysis on the quantified standard bands was, in turn, used to infer the amount of ß-CATENIN-1 in the experimental samples on the same blots. The sample concentration was calculated by normalizing the measured ß-CATENIN-1 amount to the loaded sample volume (protein lysate). Unless noted otherwise, each experiment was carried out in three biological replicates (independent lysate preparations), each comprising four technical replicates (independent gels/blots of the same lysate sample).

## Histological analysis of yolk glands

Specimens were first relaxed with a 1:3,000 dilution of linalool (Sigma, L2602)[92] and subsequently killed and fixed with Bouin's solution[93] for 12 h, transferred to 70% ethanol and cleared in xylene before paraffin embedding. Sectioning and staining were conducted by the Dresden Concept Histology Facility. The 5 µm transverse sections were attached to glass slides, stained in Mallory-Casson and mounted in DPX. All wide-field images were generated on an ZEISS Axio Scan.Z1 slide scanner. The percentage area occupied by yolk gland tissue in images of five prepharyngeal cross-sections was outlined manually on the basis of the strongly contrasting yolk granules and quantified using the software Fiji[94].

## Statistics and reproducibility

For the quantification of head regeneration in wild-type animals (Fig. 2a,d,e and Supplementary Fig. 2a,b), a representative image of the indicated phenotype is shown where applicable; the number pairs illustrate the frequency of the shown phenotype (first number) within the total experimental cohort (second number), yielding the relative frequency of the phenotype. The experiment was repeated generally once but twice for *C. pinguis* in Fig. 2 and *D. japonica* in Supplementary Fig. 2b. The images and the numbers shown are representative of a single experiment in all cases.

For the quantification of head regeneration in RNAi-treated animals in Fig. 4c, the experiment was repeated three times for *C. pinguis* and *P. tenuis*, twice for *C. robusta* and once for *P. torva* and *B. candida*; the number pairs illustrate relative phenotype frequency as defined above.

For whole-mount in situ hybridizations (Fig. 5c,e,f,l and Supplementary Fig. 5b,c,d,e,g), a representative image of the indicated phenotype is shown; the number pairs illustrate relative phenotype frequency as defined above.

In situ hybridizations were generally repeated once but twice for Fig. 5f,l, Supplementary Fig. 5d,e and for the sexual strain of *Smed* in Supplementary Fig. 5g. In the last case, data from both replicates are shown.

The SDS−PAGE gels in Supplementary Fig. 5f and in Fig. 5h, the fluorometric western plots in Figs. 4a and 5i,j and Supplementary Fig. 6a, correspond to a single experiment. The quantitative Western blotting in Fig. 6g corresponds to a single experiment with four technical replicates.

The association between regeneration ability, reproduction mode and ß-CATENIN-1 abundance (Fig. 6f,i) was tested using linear mixed-effects models. First, the measured ß-CATENIN-1 amounts in tail tip lysates were $\log_{10}$-transformed and inspected for approximately normal distribution. Then a linear mixed model was fit using the lmer function in the R package lme4 (v.1.1-30) in R (v.4.1.2), with regeneration ability or sexual system as the fixed effect and species identity as a random effect: $y = x + (1 \mid \text{species})$. For significance testing, the degrees of freedom were approximated using Satterthwaite's method as implemented in the R package lmerTest (v.3.1-3). The same method was used to evaluate the relationship between regeneration ability and yolk abundance.

## Reporting summary

Further information on research design is available in the Nature Portfolio Reporting Summary linked to this article.

## Data availability

All newly reported gene sequences have been submitted to GenBank and are available under the accession numbers listed in Supplementary Table 6. All transcriptome assemblies are publicly available at Zenodo (https://doi.org/10.5281/zenodo.8301321) and will be made available in PlanMine in the future. All raw RNA sequencing reads are available at the SRA under the BioProject accession PRJNA1011852. Previously published transcriptomes are described in the following references: *D. lacteum*[25]; *P. fluviatilis*[26]; *P. vittatus* and *K.* cf. *amphipodicola*[40]; *S. polychroa, D. japonica, P. nigra, P. tenuis, S. mediterranea*[46]; *M. lignano*[95]; *G. dorotocephala, Ph. gracilis, Ph. morgani*[96]; and the BioProject accession PRJDB1529 for *B. annandalei*.

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

## Acknowledgements

We thank H. Bilandžija, M. Handberg-Thorsager, S. D'Aniello, T. Stückemann, O. Timoshkin and T. Boothe for fieldwork support. We thank L. Drees, L. Gasiorowski, P. Ditte, F. Ficze, and M. Brockmeyer for support with experimental work. We thank H. Andreas, R. Kluivert, J. Krull, T. Schubert and the MPI-NAT animal services staff for worm care and species collection maintenance. We thank J.-H. Lee and I. Henry for support with phylogenetic analyses and T. Boothe for imaging support. We thank the following facilities for their support: the Center for Molecular and Cellular Bioengineering (CMCB) Histology Facility, the Dresden Concept Genome Center and the CMCB Light Microscopy Facility. We thank the following MPI-CBG facilities for their support: the Protein Expression and Purification Facility, the Antibody Facility and the Mass Spectrometry Facility. We thank Rink laboratory members for lively discussions. We acknowledge Instituto Florestal and the administration of the Parque Estadual do Juquery for licensing sampling and help with fieldwork (002947/2021-83). This project received funding from the European Research Council under the European Union's Horizon 2020 research and innovation programme (grant agreement no. 649024), from the German Research Foundation (project RI 2449/51), from the Behrens-Weise Foundation and from the Max Planck Society funding (J.C.R.). J.E.J.R. acknowledges funding from the NHMRC Investigator grant (no. 1177305) and the Sir Zelman Cowen Foundation.

## Author contributions

J.C.R. and M.V.-F. conceptualized the study. M.V.-F., J.C.R., J.C., C.E.P., B.E., M.A.G., J.E.J.R., H.T.-K.V. and F.C. collected samples. M.V.-F., M.I., J.C., S.K., M.A.G. and A.L.G. performed the experiments. A.R., J.N.B., F.S. and C.B. performed the data analysis for the phylogeny. M.V.-F. and J.C.R. wrote the manuscript and revised the manuscript text.

## Funding

## Competing interests

## Additional information

**Correspondence and requests for materials** should be addressed to Miquel Vila-Farré or Jochen C. Rink.

[1]Department of Tissue Dynamics and Regeneration, Max Planck Institute for Multidisciplinary Sciences, Göttingen, Germany. [2]Max Planck Institute of Molecular Cell Biology and Genetics, Dresden, Germany. [3]Animal Evolution and Biodiversity, Georg-August-Universität Göttingen, Göttingen, Germany. [4]Cardio-CARE, Medizincampus Davos, Davos, Switzerland. [5]Department of Zoology, University of Innsbruck, Innsbruck, Austria. [6]European Molecular Biology Laboratory, Heidelberg, Germany. [7]Department of Zoology & Animal Cell Biology, University of the Basque Country (UPV/EHU), Leioa, Spain. [8]Laboratório de Ecologia e Evolução. Escola de Artes, Ciências e Humanidades, Universidade de São Paulo, São Paulo, Brazil. [9]Gene and Stem Cell Therapy Program Centenary Institute, Camperdown, New South Wales, Australia. [10]Faculty of Medicine and Health, University of Sydney, Sydney, New South Wales, Australia. [11]Cell & Molecular Therapies, Royal Prince Alfred Hospital, Camperdown, New South Wales, Australia. [12]These authors contributed equally: Andrei Rozanski, Mario Ivanković, James Cleland. ✉e-mail: mvilafa@mpinat.mpg.de; jochen.rink@mpinat.mpg.de

Jochen C. Rink

# Reporting Summary

## Statistics

For all statistical analyses, confirm that the following items are present in the figure legend, table legend, main text, or Methods section.

| n/a | Confirmed | |
|---|---|---|
| ☐ | ☒ | The exact sample size (*n*) for each experimental group/condition, given as a discrete number and unit of measurement |
| ☐ | ☒ | A statement on whether measurements were taken from distinct samples or whether the same sample was measured repeatedly |
| ☐ | ☒ | The statistical test(s) used AND whether they are one- or two-sided *Only common tests should be described solely by name; describe more complex techniques in the Methods section.* |
| ☒ | ☐ | A description of all covariates tested |
| ☐ | ☒ | A description of any assumptions or corrections, such as tests of normality and adjustment for multiple comparisons |
| ☐ | ☒ | A full description of the statistical parameters including central tendency (e.g. means) or other basic estimates (e.g. regression coefficient) AND variation (e.g. standard deviation) or associated estimates of uncertainty (e.g. confidence intervals) |
| ☐ | ☒ | For null hypothesis testing, the test statistic (e.g. *F*, *t*, *r*) with confidence intervals, effect sizes, degrees of freedom and *P* value noted *Give P values as exact values whenever suitable.* |
| ☒ | ☐ | For Bayesian analysis, information on the choice of priors and Markov chain Monte Carlo settings |
| ☒ | ☐ | For hierarchical and complex designs, identification of the appropriate level for tests and full reporting of outcomes |
| ☒ | ☐ | Estimates of effect sizes (e.g. Cohen's *d*, Pearson's *r*), indicating how they were calculated |

*Our web collection on statistics for biologists contains articles on many of the points above.*

## Software and code

Policy information about availability of computer code

| Data collection | All the programs used for collecting data are reported in the Methods section of the paper and/or in the figures. |
|---|---|
| Data analysis | All the programs used for data analysis are reported in the Methods section of the paper and/or in the figures. |

For manuscripts utilizing custom algorithms or software that are central to the research but not yet described in published literature, software must be made available to editors and reviewers. We strongly encourage code deposition in a community repository (e.g. GitHub). See the Nature Portfolio guidelines for submitting code & software for further information.

## Data

Policy information about availability of data

All manuscripts must include a data availability statement. This statement should provide the following information, where applicable:
- Accession codes, unique identifiers, or web links for publicly available datasets
- A description of any restrictions on data availability
- For clinical datasets or third party data, please ensure that the statement adheres to our policy

All raw sequencing data generated furing the course of this project will be made publicly available in an online repository (Genbank) before publication. Theassembled transcriptomes will be additionally made available via the PlanMine website (https://planmine.mpibpc.mpg.de/planmine/begin.do).

# Research involving human participants, their data, or biological material

Policy information about studies with human participants or human data. See also policy information about sex, gender (identity/presentation), and sexual orientation and race, ethnicity and racism.

| | |
|---|---|
| Reporting on sex and gender | N.A. |
| Reporting on race, ethnicity, or other socially relevant groupings | N.A. |
| Population characteristics | N.A. |
| Recruitment | N.A. |
| Ethics oversight | N.A. |

Note that full information on the approval of the study protocol must also be provided in the manuscript.

# Field-specific reporting

Please select the one below that is the best fit for your research. If you are not sure, read the appropriate sections before making your selection.

☐ Life sciences    ☐ Behavioural & social sciences    ☒ Ecological, evolutionary & environmental sciences

For a reference copy of the document with all sections, see nature.com/documents/nr-reporting-summary-flat.pdf

# Ecological, evolutionary & environmental sciences study design

All studies must disclose on these points even when the disclosure is negative.

| | |
|---|---|
| Study description | Our study systematically explores the gain and loss of head regeneration in planarians by means of a live collection of more than 40 planarian species. Our experimental approach includes the quantification of species-specific head regeneration abilities, WISH (whole mount in situ hybridization), gene function analysis by RNAi, histology and quantitative Western Blotting to measure protein abundance. Experiments included a minimum of three individuals/species, depending on experimental requirements and animal availability. The association between regeneration ability, reproduction mode and ß-CATENIN-1 abundance was tested using linear mixed-effects models. |
| Research sample | Our study involved a broad range of planarian species that represent the major branches of planarian phylogeny. Our species collection was established by dedicated field sampling in known hotspots of planarian diversity. Between 10 to 100 individuals/species were collected at a field location and subsequently established as a laboratory population. Our study assumes those individuals and their offspring as representatives of the species. The transcriptome assemblies are based on RNA extracted from 3 to 6 individuals of field-collected populations. To measure head-regeneration abilities, we applied the standardised amputation-regeneration paradigms described in the text. All stainings were performed on individuals derived from progeny of field-collected populations except for S. mediterranea, of which we maintain clonal laboratory populations. Planarians are not subject to ethics approval in the EU. |
| Sampling strategy | All laboratory experiments were conducted with sufficient sample sizes for the specific purposes of the experiment, in line with the standards of the planarian research community. |
| Data collection | All data collection contributors are among the authors of the paper (please see the author's contribution statement). The data collection procedure for specific experiments is detailed in the Methods section. |
| Timing and spatial scale | Field collections were performed mainly before 2015. Experiments on the laboratory strains derived from field-collected specimens were performed between 2014-2021. |
| Data exclusions | In the case of Quantitative Western blotting (Fig. 4a and b, 5d, 5h to k, 6b, 6h), some replicates were excluded for obvious technical failures (e.g., insufficient band resolution due to DNA contamination). In the histology quantifications of yolk gland cross-sectional areas in Figures 6e and 6f, some replicates were excluded due to the absence of yolk glands in specific specimens. For this reason, we state in the text that " This tendency was statistically significant despite substantial inter-animal variations in the yolk content in some species (e.g., P. tenuis, P. torva or Camerata robusta) (Fig. 6f), which may reflect non-synchronous reproduction cycles under our laboratory culture conditions". |
| Reproducibility | The number of technical and biological replicates is clearly stated in the text. |
| Randomization | Specimens were selected randomly from laboratory populations. Where necessary, sexually mature individuals were preselected based on the presence or absence of a gonopore. |

| Blinding | Blinding was not necessary due to the use of either unambiguous assays (e.g., presence-absence of eyes as an indicator of head regeneration) or quantitative assays (e.g., fluorescence quantification of WB band intensities). |

Did the study involve field work? ☒ Yes ☐ No

## Field work, collection and transport

| Field conditions | We conducted field sampling in multiple localities worldwide (see map in Fig. 1a). Sampling was performed as described in the manuscript. Seasonal variations in species abundance were not part of this study and were not analysed. |
| Location | A map with the sampled locations is shown in Fig. 1a. More precise data will be provided via planmine. |
| Access & import/export | Our collection is compliant with the Nagoya agreement (ascertained in collaboration with the German Nagoya Hub). Our fieldwork generally involved local researchers that are more familiar with the local regulations (see author contributions). Field sampling in Australia was performed under the Australian permit to take wildlife for scientific purposes No: 12239. Field sampling in Brazil was performed under the Brazilian permit for sampling No:02947. |
| Disturbance | Our sampling approach is minimally invasive per se, since we manually collect a small number of animals from submerged stones or other objects. We further minimise environmental impact by returning stones to their original position in the river bed and by avoiding sensitive habitats altogether. |

# Reporting for specific materials, systems and methods

We require information from authors about some types of materials, experimental systems and methods used in many studies. Here, indicate whether each material, system or method listed is relevant to your study. If you are not sure if a list item applies to your research, read the appropriate section before selecting a response.

## Materials & experimental systems

| n/a | Involved in the study |
|---|---|
| ☐ | ☒ Antibodies |
| ☒ | ☐ Eukaryotic cell lines |
| ☒ | ☐ Palaeontology and archaeology |
| ☐ | ☒ Animals and other organisms |
| ☒ | ☐ Clinical data |
| ☒ | ☐ Dual use research of concern |
| ☒ | ☐ Plants |

## Methods

| n/a | Involved in the study |
|---|---|
| ☒ | ☐ ChIP-seq |
| ☒ | ☐ Flow cytometry |
| ☒ | ☐ MRI-based neuroimaging |

## Antibodies

| Antibodies used | This study involved five antibodies. The penta-His, H3 and H3P antibodies are commercial. Our anti-Smed-b-CATENIN-1 monoclonal antibody clone G78 was described previously (publication reference provided in the manuscript). The methods used for producing and testing the custom-raised anti-FERRITIN antibody, clone EO95, are described in the manuscript. Both antibodies are available upon request from the corresponding authors. |
| Validation | The methods used for validating the clone EO95 are described in the manuscript and involved the loss of the band upon RNAi-mediated knock-down of the epitope. |

## Animals and other research organisms

Policy information about studies involving animals; ARRIVE guidelines recommended for reporting animal research, and Sex and Gender in Research

| Laboratory animals | We used two laboratory strains of Schmidtea mediterranea: the clonal CIW4 strain (asexual) and the inbred strain S2 (sexual). Both are established laboratory lines that are studied in many laboratories worldwide. Additionally, we also use laboratory strain of Dugesia japonica and Dugesia tahitiense. |
| Wild animals | The field-collected species and collection techniques are detailed in the manuscript. Briefly, animals were shipped on wet ice in 50 ml falcon tubes, at a density of ~20 individuals per tube, with 45 ml of water from the collection locality. The combinatorial method detailed in the manuscript was used to establish suitable laboratory culture conditions. Experiments were mainly performed on species from which we obtained stable laboratory cultures. |
| Reporting on sex | Sex considerations generally do not apply, as the planarians used in this study are either hermaphrodites or asexual. The exceptions are Sabussowia dioica (species with separate sexes) and Hymanella retenuova (likely a sequential hermaphrodite). Sex was not taken |

into account when producing transcriptomes for those species or for the head regeneration assay in H. retenuova due to the difficulty of sexing live animals. The degree of sexual maturity of hermaphrodites, which is relevant for some of our experiments, was assessed by the presence of a gonopore. Conversely, the absence of the gonopore was used to ascertain the asexuality of specific strains.

Field-collected samples | General culture parameters for field-collected species are described in the manuscript, as are typical fixation and lysis protocols.

Ethics oversight | N.A. -Planarians are not subject to ethical oversight regulations.

Note that full information on the approval of the study protocol must also be provided in the manuscript.

