## [Peer Review File · Nature Ecology & Evolution]

Peer Review Information

Journal: Nature Ecology & Evolution

Manuscript Title: Evolutionary dynamics of whole-body regeneration across planarian flatworms

Corresponding author name(s): Miquel Vila-Farré, Jochen C. Rink

Editorial Notes:

Reviewer Comments & Decisions:

Decision Letter, initial version:

8th February 2023

Dear Miquel,

Your manuscript entitled "Probing the evolutionary dynamics of whole-body regeneration within planarian flatworms" has now been seen by three reviewers, whose comments are attached. The reviewers have raised a number of concerns which will need to be addressed before we can offer publication in Nature Ecology & Evolution. We will therefore need to see your responses to the criticisms raised and to some editorial concerns, along with a revised manuscript, before we can reach a final decision regarding publication.

We therefore invite you to revise your manuscript taking into account all reviewer and editor comments. Please highlight all changes in the manuscript text file in Microsoft Word format.

* If you have not done so already please begin to revise your manuscript so that it conforms to our Article format instructions at <http://www.nature.com/natecolevol/info/final-submission>. Refer also to any guidelines provided in this letter.

[REDACTED]

2Note: This URL links to your confidential home page and associated information about manuscripts you may have submitted, or that you are reviewing for us. If you wish to forward this email to co-authors, please delete the link to your homepage.

Nature Ecology & Evolution is committed to improving transparency in authorship. As part of our efforts in this direction, we are now requesting that all authors identified as 'corresponding author' on published papers create and link their Open Researcher and Contributor Identifier (ORCID) with their account on the Manuscript Tracking System (MTS), prior to acceptance. ORCID helps the scientific community achieve unambiguous attribution of all scholarly contributions. You can create and link your ORCID from the home page of the MTS by clicking on 'Modify my Springer Nature account'. For more information please visit www.springernature.com/orcid.

[REDACTED]

Reviewer expertise:

Reviewer #1: molecular mechanisms of planarian regeneration

Reviewer #2: molecular mechanisms of regeneration

Reviewer #3: evolution of animal genomes, comparative omics and phylogenetics

Reviewers' comments:

Reviewer #1 (Remarks to the Author):

This work characterizes head regeneration capability across a range of flatworm species and provides functional experiments to indicate a widespread ability of canonical Wnt perturbation at bypassing head regeneration deficiency. The overall scale of work is truly impressive, and the planarian species resource presented here generates a treasure trove of unique observations to study how regeneration ability has been tuned through evolution. There are a number of unique and interesting accessory observations as well, for example, that *Dugesia japonica* has a latent tendency for posterior head

2regeneration, and that *D. Sicula* undergoes bifurcated regenerated after lateral amputations. These and the other observations of the paper help open a very wide area for future investigations. In addition, the authors provide highly convincing data to show that head regeneration has been lost/gained several times in the flatworm lineage, and also that beta-catenin RNAi is able to rescue head regeneration from species which independently would have evolved head regeneration deficiency. These major findings make are highly significant set of statements regarding evolution of regeneration abilities.

The study then explores a possible negative relationship between sexual reproduction and head regeneration ability because of a correlation between strong head regeneration and fissiparous lifestyle using asexual reproduction. This is an interesting hypothesis to examine, and the authors find that Wnt signaling components are expressed in the Smed reproductive system, and that beta-catenin RNAi leads to reduced expression of reproductive yolk gland factors ferritin and surfactant-b. Though the exact nature of Wnt/beta-catenin signals in control of the Smed reproductive system is not entirely resolved here, the findings argue for the use of this pathway in activities critical for planarian sexual reproduction. By measuring relative yolk gland size histologically and performing quantitative western blots, the study then explores the relationship across species between yolk abundance, overall total animal levels of beta-catenin protein, and head regeneration ability. The study suggests a trend in which greater yolk abundance somewhat, but not precisely, correlates with lower head regeneration ability. Likewise, the authors suggest a correlation between total animal levels of beta-catenin protein and lower head regeneration ability, although the meta-analysis combining all biological replicates across species failed to find a statistically significant difference across regeneration extents.

Overall this is a comprehensive and I believe landmark advance to the field through resolving head regeneration abilities across many newly characterized species, as well as finding a broad ability for beta-catenin inhibition to enhance or restore this ability. The work to explore relationships between beta-catenin levels, allocation of resources to reproduction versus regeneration is an important step forward, even if the conclusions at present are less clear or universal. Based on the overall scope and findings, the study it would be of interest to a broad audience, however here are several critical aspects to the work that must first be addressed.

1. The authors propose a model in which Wnt from the reproductive system interferes with head regeneration. However, the data from Smeds already argues against this concept, because sexual Smeds regenerate their heads as well as asexual Smeds, yet have higher beta-catenin levels across their body which are also ophis-dependent. I assume, but the authors have not shown, that ophis-inhibited animals do not have any more rapid or greater ability of head regeneration. The original reporting of this gene did not note such defects, so I think it is unlikely that they have enlarged heads or generate ectopic heads. On the other hand, if ophis inhibition did increase head regeneration in some perhaps subtle way, this would provide a straightforward way to test the very interesting idea that the sexual reproduction program indeed dampens regeneration ability. Perhaps an assay that measures head regeneration speed/extent with higher sensitivity could find such a relationship. This analysis would allow further tests to functionally implicate Wnt from the reproductive system in this process, by determining whether ophis inhibition might then eliminate the ability for beta-catenin RNAi to further enhance head regeneration or cause anteriorization. Alternatively, perhaps ophis and APC

3would genetically interact in sexual animals for control of head-tail regeneration defects, which could also support such a functional relationship. Also it seems that the *Ptor*, *Dlac* and *Pten* genomes have factors that are reciprocal blast best matches to *Smed-ophis* (*dd_Ptor_v3_1798_1_1*, *dd_Dlac_v9_189091_0_1*, *dd_Pten_v3_33198_1_1*) so these seem like excellent candidates that could ultimately allow testing whether beyond *Smed* there is the negative relationship between Wnt/sexual reproduction versus head regeneration as proposed here. However, at present, given the strong likelihood from *Smeds* that the ophis-dependent beta-catenin portion does not influence head regeneration at all, the burden of proof rests on the authors to show functional support for the idea that Wnt levels from the sexual reproduction system participate can interfere with head regeneration in any flatworm species.

2. Based on data presented here, there is an ambiguity in the interpretation of whole tissue or whole animal beta-catenin levels that affects the tests of the main hypothesis. How much of the beta-catenin measured here is relevant for head/tail regeneration? As the authors nicely show, several Wnt pathway components including feedback regulators such as axin and notum, have expression in the reproductive system, and therefore reproductive system components are likely themselves sources of local Wnt pathway activation. Additionally, sexual *Smeds* undergo head regeneration seemingly equally as well as the asexual *Smeds*. Therefore, the simplest interpretation of the western blotting results in Fig 5D is that a substantial portion of the detected animal-wide beta-catenin gradient in sexual animals is likely to be due to beta-catenin protein acting locally within the reproductive system, rather than as suggested that sexual animals have an intrinsically higher gradient of beta-catenin that is actively functional in head-tail determination (for example, in muscle). This first more likely conclusion would also affect the interpretation of western blotting measurements of whole-animal levels of beta-catenin in non-regenerative species. If much of the beta-catenin measured across species is simply a readout of amounts of reproductive tissue present in animals, it is less clearly a relevant or separate metric for predicting regeneration ability according to the overall hypothesis. This issue affects Fig 6g-i and the analysis of beta-catenin levels across species, because from Figure 5d, at least ~2/3 of the beta-catenin protein signal from tails of the sexual *Smeds* is lost in ophis RNAi and therefore not relevant to head-tail fating (because ophis inhibited animals have not been noted to have problems with head/tail identity). Because the fraction of beta-catenin used for tail patterning might differ across species, there is a concern that western blotting might not give sufficiently resolved information to weigh in strongly on the hypothesis.

3. I must say that the manuscript really inadequately references the very relevant and abundant scope of literature revealing the roles for Wnt signaling in head and tail regeneration in planarians. The authors sufficiently reference their own work but unfortunately not important work from the rest of the field that also lays the foundation leading to this study and helps to interpret its findings. For example, many studies not cited here have observed suppression of headless phenotypes by beta-catenin RNAi, which are relevant for interpreting the results from similar approaches used here across flatworm species. Many other studies not cited here have reported on the mechanism of injury-induced Wnt activation or inhibition as well as uses for systemic Wnt components in control of head/tail pattern which would be relevant for considering the possibly many ways that species could evolve higher Wnt signaling (which signals and at what steps) that might block head regeneration. Other studies have also examined beta-catenin and the reproductive system, with one study even showing beta-catenin immunostaining in the reproductive system of *S. polychroa* (Sureda-Gómez

42016). The neglect of abundant literature in the field gives the unfortunate impression that the authors have not deeply considered their study in light of other findings.

4. The monoclonal antibodies described here, both the antibody newly reported for yolk proteins and the antibody described in 2017 for beta-catenin, should be deposited to a commercial or public hybridoma bank. This would ensure the long-term durability of the reagents for the field and also critically to meet rigor and reproducibility standards that allow findings of the work to be independently verified by other labs.

5. More information about production of the EO95 antibody is needed in the methods, rather than citing prior work with the antibody core facility.

6. Figure 6f and 6i need y-axis labels and red lines defined (are these means, medians, etc).

7. On p. 14, the authors refer to data not shown of a head-tail beta-catenin protein gradient present in most of the species investigated here. The authors either need to show the data or remove the statement, as such statements cannot be evaluated by peer review.

8. I'm not sure that the data in Fig 5 really show a gradient of beta-catenin protein throughout the body of the asexual Smed strain. When a statistical test is used to determine which adjacent regions have more or less beta-catenin, which regions show a statistically significant difference? The overlapping error bars make it difficult to be sure, but it seems the drop between region 6 to 5 is substantial and I think likely to be significant, but for example the levels across regions 1 through 4 do not seem to be very different from each other. It seems more accurate to call this a gradient within the tail rather than an animal-wide gradient.

9. It is interesting that the beta-catenin and APC RNAi was not successful in *P. Tenuis*. Is it possible that the particular dsRNAs chosen for these experiments might simply be ineffective, or that the target proteins have a particularly long half-life, or tissues themselves undergo very slow turnover etc, rather than indicating a general lack of RNAi in that organism? I agree it is suggestive of lack of useful RNAi in general for that organism, but the claim is based on negative data.

10. Typo p6 "iapable"

11. The statement at the top of p.10 "these results further imply that head regeneration defects may be associated with a functional excess of Wnt pathway activity..." is a hypothesis, but it seems the authors should consider at this point the alternative of bypass suppression and present the subsequent analysis more explicitly as a test of these two possibilities. Or alternatively, that they chose to investigate this particular possibility.

12. Please modify the language used to describe whether the model has been supported versus refuted. The statement describing the statistical test in beta-catenin levels reflects a bias in study design. It reads : "The statistical analysis of tail tip...indicated a tendency towards higher beta-catenin levels...yet without reaching statistical significance." Yet, what kind of outcome other than what was observed would have been able to disprove the hypothesized association? This text should be revised

5to say "but such an association was not statistically significant." There is a similar issue with the description of the results that "failed to unequivocally support" rather than, as is the case, simply failing to provide support for the hypothesis.

Similarly, I can see that the authors try to temper their conclusions about the Wnt/reproduction model in the abstract "Although initial quantitative comparisons...confirm some...they also highlight diversification..." However, based on the considerations from points 1 and 2, which predictions of the Wnt model do the authors claim are confirmed? There is not yet evidence that reproductive systems functionally impair planarian head regeneration, and as argued above there is not yet clear evidence that species with lower head regeneration ability have greater Wnt/beta-catenin signaling levels that are participating in head/tail regeneration.

Perhaps it would be better to state that quantitative methods were applied to attempt to test this hypothesis but it remains unresolved.

Reviewer #2 (Remarks to the Author):

A long-standing question in biology is why only certain animals can regenerate, while many others cannot. In this work, the authors aim to address this question by comparing head regeneration among ~40 species of planarian flatworms. Using a quantitative assay for head regeneration, the authors classified planarians into three groups - robust regeneration, restricted regeneration, and poor regeneration. Based on this classification, newly generated phylogenetic trees from transcriptomes, and other published reports, they predict that the likeliest ancestral state was 'poor regeneration'. Additionally, they predict that the capability for head regeneration has been independently gained and reduced at least 5 and 3 times respectively.

From previous studies in three different planarian species, we understand that the failure to regenerate heads is due to mis-regulated WNT/ β -catenin signaling. The authors find that this model is widely conserved as evidenced from the β -catenin(RNAi) experiments in 2 species from restricted regeneration group and 2 species from poor regeneration group. The frequency of head rescue correlated well with the RNAi knockdown efficiencies, which was measured using the *Schmidtea mediterranea* β -catenin antibody. From these experiments, the authors conclude that the WNT/ β -catenin signaling is a 'putative hot spot in the evolution of planarian head regeneration defects'.

As misregulation of the WNT/ β -catenin signaling leads to head regeneration defects, and poor regeneration correlates with egg-laying, the authors hypothesize that there exists a pleiotropic role of the WNT pathway in egg production. In sexual *Schmidtea mediterranea* animals they observe that many genes of the WNT pathway are expressed in the reproductive system and is required for maintenance of yolk glands. Based on these experiments, the authors propose that there exists a tradeoff between regeneration and egg-laying - because yolk production requires enhanced WNT/ β -catenin signaling and that inhibits head regeneration. According to the model, sexually reproducing animals are expected to express higher levels of β -catenin, but that is not the case. Additionally, WNT/ β -catenin signaling is not always required for maintaining yolk glands as it is in sexual *S. mediterranea* animals. Based on these experiments, the authors conclude that the changes in the

6WNT/ β -catenin signaling may not explain all the differences, but the working model of trade-off between regeneration and reproduction provides a working hypothesis for future work.

This work conclusively shows that increased WNT/ β -catenin signaling is the cause for reduced/failed head regeneration in multiple species. Additionally, the authors have collected ~40 species of planarians and have developed laboratory husbandry protocols which is a great resource for the community to carry out more comparative studies.

However, there are two main areas which need to be clarified –

1. A major portion of the work focusses on building the planarian flatworm phylogeny and predicting the ancestral state of regeneration. The analyses described in the results are obtuse and fails to build a clear picture about the evolutionary history of regeneration. It is unclear as to how the head regeneration data was used in ancestral state reconstruction. These efforts improve the phylogenetic trees for planarian flatworm but as presented, it is not certain that they provide insights on the evolution of head regeneration.

2. The tradeoff between regeneration and reproduction is one of the possibilities driving regeneration, but there can be others like predation which need to be considered and discussed as part of working models/hypotheses. Secondly, from Fig S5.a, we can observe that animals from all the three groups of head regeneration occupy both lentic and lotic habitats. This makes the correlation between mode of reproduction and habitat of the animals a bit stretched. The authors also suggest that asexual reproduction might be able to reestablish a colony at a faster pace compared to sexual animals, which would be a necessity in ephemeral ponds. However, we do not know if this is true as animals need to grow to a certain size before they can fission and the fission fragment is highly susceptible to predation. Thus, the selection pressure for regeneration can be driven by predation in the various habitats. It may be hard to measure predation levels in the wild, but I think it should be discussed in the text.

Questions

1. Figure 3: Phylogenetic trees are hard to read, and they are of low resolution.
2. *P. tenuis* was not susceptible to β -catenin(RNAi) is cool. Are there duplicate genes for β -catenin in this species?
3. In these β -catenin(RNAi) experiments, did the animals form a head at the posterior blastema in the different species tested?
4. In Fig 4C, does the control *C. robusta* animal have 2 pharynges?
5. In Fig 4C, please indicate number of the animals with the shown phenotype rather than number of animals with no head.
6. Is WNT/ β -catenin signaling known to be required for yolk production in other animals? Is that a conserved function across metazoans?
7. Is the expression of WNT pathway members in the reproductive system conserved across multiple planarian species?

Reviewer #3 (Remarks to the Author):

7Schmidtea planarians have been a hotspot for regeneration study, given their high capabilities of regenerating any missing body parts. This capability is because Schmidtea has a large pool of adult pluripotent stem cells (neoblasts) and robust positional information. In this study, Vila-Farré and Rink expanded and explored the evolutionary origins of regeneration mechanisms in Platyhelminthes, a clade of flatworms.

People often neglect the fact that platyhelminths are spiralian with high diversity. As a result, the conceptual framework of neoblasts and regeneration models in Schmidtea planarians have been applied in various studies without recognizing the fundamental difference and diversity among animal groups. An unbiased evolutionary perspective is lacking partly because of this fixed perspective. This study is a pioneering effort to close this knowledge gap. And indeed, some line of evidence shows that platyhelminths are fast evolving and derive when compared with other relatively conserved members in the clade Spiralia. For example, genomic evidence shows that some platyhelminths experienced substantial changes, including expansion of genome sizes and gene loss. They also possess quite diverse body plans. It is thus still not so well understood how the regeneration capabilities evolved.

Through an extensive effort of field collection and establishing a lab culture system for more than 40 planarian species, the authors systematically and quantitatively examined the regenerative abilities across a group of flatworms in Tricladida. Using transcriptomic analysis together with anterior-to-posterior serial cutting and RNAi perturbation, the authors classified planarians into three major groups, i.e., robust (A), restricted (B), and poor (C) head regeneration. They then explored the correlation between canonical Wnt signaling and yolk production using a biochemical approach (i.e., quantification of the protein expression level of beta-Catenin and ferritin using custom antibodies). Finally, the author proposed a model to answer the evolutionary scenarios of regeneration.

Overall, the authors presented an interesting and well-elaborated study. The presentation of data is of high quality, and the topics are of broad interest to the journal readers. And the authors also proposed some working hypotheses for future studies. I mainly have comments on evolutionary scenarios (i.e., a more general scheme when considering evolution through gene evolution), signaling components, and the role of neoblasts.

Major comments

1. One major factor in probing the molecular basis of evolutionary dynamics is the gene content. Since the authors focused on canonical Wnt signaling and already have high-quality transcriptome data, it would be great to see the distribution of Wnt signaling components across planarian species. For example, is there any Wnt ligand expansion or loss in particular lineages or regeneration groups? We know that Wnt3 is absent in all examined spiralian genomes. And in some platyhelminths, Wnt6 to Wnt10, Wnt16, and WntA are all lost. Would it be possible that some Wnt ligands have not been lost in planarian species with poor regeneration? In other words, would the loss of Wnt signaling ligands be a key evolutionary transition to whole-body regeneration?

2. A similar line to the first comment, Wnt signaling can be regulated through different ligands as well as antagonists, such as the anterior marker gene, notum. Did the author check the expression of Wnt

8antagonists? For example, at the transcriptome level, do some flatworms have more Wnt antagonists than others? How are these antagonists expressed in relation to their body or organ structures?

3. It is well-known that Wnt signals (i.e., Wnt ligands) are carried in muscle cells. Thus, muscle cells are not only contractile but also a coordinate system with positional information. It would be interesting to see how major Wnt ligand orthologs are expressed in the flatworm collection. And how are these muscle cells correlated to the reproductive system?

4. The model that the author proposed attempts to explain the tradeoff when asexual and sexual reproduction modes are under natural selection, depending on the ecological niche. I am not sure if fission (spontaneous "pinch off") and regeneration (damage and then wound responses) are the exact cellular and molecular processes. It seems that the authors consider them the same thing. For the flatworms capable of fission, do they also have wounded tissues or wound responses during fissiparous reproduction? Given wounding seems to be an initial signal for triggering damage-induced regeneration. Would it be possible to have three different reproduction modes instead of just two (i.e., damage-induced regeneration, fission, and egg-laying)?

5. When reconstructing the ancestral state, did the authors consider the synapomorphy of flatworms? i.e., which clade shares the derived trait of regeneration? This is not very clear in the current manuscript writing.

6. A fundamental component largely ignored in this study is the neoblast, the source pluripotent stem cells that provide all differentiating cell types. In some systems, the distribution of neoblasts along the body axis is correlated to the regeneration capabilities of particular body parts. Did the author consider this point? And how do the authors combine this consideration into their evolution model? Also, in case the species have sexual reproduction, how can the current model explain the dynamics between the behavior of somatic pluripotent stem cells and germline stem cells?

Minor comments:

1. Figure 1: It would be easier for readers to understand different regeneration groups if the authors could color-code the dots in panel a and provide color-coded dots in panels b–d.

2. Figure 5b: It is recommended to add a tree to the clades on the left. It would also be helpful to highlight group A and fissiparous reproduction. Overall, the font size in Figure 5 is too small to read.

3. In the Introduction, please explain canonical Wnt signaling and how it is linked to beta-catenin to help readers without this background.

4. Replace *Smed* with *S. mediterranea* to have consistency across all flatworm species.

5. Line 76: A/P axis, spell it out (e.g., anterior/posterior axis).

6. Line 119: Write out MPI-NAT.

7. Line 165: Dugesia sicula -> D. sicula

8. Line 166: Dugesia japonica -> D. japonica

9. Line 179: iapable -> incapable

10. Avoid using the acronym "RS" for the "reproductive system." This adds unnecessary difficulty to the reading.

11. Line 376: Dugesia ryukyuensis -> D. ryukyuensis

*****END*****

Author Rebuttal to Initial comments

Rebuttal

We here submit the revised version of our manuscript titled “Probing the evolutionary dynamics of whole-body regeneration within planarian flatworms”.

First, we would like to thank all of our reviewers for their general appreciation of the work and constructive manuscript improvement suggestions. As detailed below, we have tried our very best to incorporate experiments and text revisions to address the salient points, and we have fine-tuned the overall message of the paper. In addition, we have substantially streamlined the text and the figure legends to more closely match the journal’s format. Overall, we are of the opinion that the manuscript has improved substantially and we hope that our reviewers agree.

Reviewer #1 (Remarks to the author):

...The overall scale of work is truly impressive, and the planarian species resource presented here generates a treasure trove of unique observations to study how regeneration ability has been tuned through evolution. ... These major findings make are highly significant set of statements regarding evolution of regeneration abilities.

10Response: Thank you!

...

Overall this is a comprehensive and I believe landmark advance to the field through resolving head regeneration abilities across many newly characterized species, as well as finding a broad ability for beta-catenin inhibition to enhance or restore this ability. The work to explore relationships between beta-catenin levels, allocation of resources to reproduction versus regeneration is an important step forward, even if the conclusions at present are less clear or universal. Based on the overall scope and findings, the study it would be of interest to a broad audience, however here are several critical aspects to the work that must first be addressed.

Response: Thank you, and see below.

1. The authors propose a model in which Wnt from the reproductive system interferes with head regeneration. However, the data from *Smeds* already argues against this concept, because sexual *Smeds* regenerate their heads as well as asexual *Smeds*, yet have higher beta-catenin levels across their body which are also *ophis*-dependent.

We disagree that these observations contradict our model. We envisage the Wnt signalling pleiotropies in the reproductive system (RS) to become relevant only at extremes of reproductive performance, as explicitly stated in the text:

“...the model envisages the emergence of planarian regeneration defects via the habitat-specific selection for extreme egg capsule size or production rate and concomitant elevation of Wnt signalling levels to a point where they begin to interfere with head regeneration. The many egg-laying and whole-body regeneration competent species (e.g., the sexual *Smed* strain) are envisaged to occupy intermediate levels of the trade-off regime, i.e. intermediate Wnt signalling levels and intermediate investment in sexual reproduction that do not yet interfere with regeneration.”

Second, our model doesn't require the reproductive system (RS) to be a Wnt source. For example, the *ophis*-dependent difference in the fractional bCATENIN-1 abundance between sexual and asexual *Smed* might reflect the *response* of RS component tissues to somatic/body wall derived Wnt signals (e.g., Wnt-responsive/primed yolk or shell gland progenitors). Hence

11ophis(RNAi) might ablate bCATENIN-rich Wnt target cells but leave the somatic Wnt sources and their influence on regeneration unaffected. In contrast, strong selection pressure, e.g. for fractional neoblast progeny allocation to yolk or shell gland lineages, might increase the somatic Wnt sources beyond the normal range and thus to the species-specific regeneration defects in “specialist” species like the cited *D. lacteum* or *P. torva*. This interpretation is clearly stated in the discussion, which reads:

“Neoblasts generate and re-generate all planarian cell types¹⁴, including all constituents of the reproductive system⁷³. Hence an elevation of somatic Wnt signalling might upregulate yolk or testes formation while simultaneously interfering with head regeneration due to its dependence on Wnt inhibition. In general, our pleiotropy model assumes that regeneration and reproduction are under the influence of a common Wnt signalling source.”

I assume, but the authors have not shown, that *ophis*-inhibited animals do not have any more rapid or greater ability of head regeneration. The original reporting of this gene did not note such defects, so I think it is unlikely that they have enlarged heads or generate ectopic heads. On the other hand, if *ophis* inhibition did increase head regeneration in some perhaps subtle way, this would provide a straightforward way to test the very interesting idea that the sexual reproduction program indeed dampens regeneration ability. Perhaps an assay that measures head regeneration speed/extent with higher sensitivity could find such a relationship. This analysis would allow further tests to functionally implicate Wnt from the reproductive system in this process, by determining whether *ophis* inhibition might then eliminate the ability for beta-catenin RNAi to further enhance head regeneration or cause anteriorization.

Thanks for the suggestion. We have carried out head regeneration rate comparisons between size-matched *ophis(RNAi)* and control sexual *Smeds* (piece 2: tranverse amputation behind the photoreceptors; piece 6: tail piece). As shown below, the reviewer’s hunch was entirely correct in that regeneration rates were practically indistinguishable across *RNAi* conditions. However, as detailed above, this result neither proves nor disproves our model, which is why we opted not to include the additional data due to the severe space/word limit constraints of the journal.

Alternatively, perhaps *ophis* and APC would genetically interact in sexual animals for control of head-tail regeneration defects, which could also support such a functional relationship. Also it seems that the *Ptor*, *Dlac* and *Pten* genomes have factors that are reciprocal blast best matches to *Smed-ophis* (dd_ *Ptor*_v3_1798_1_1, dd_ *Dlac*_v9_189091_0_1, dd_ *Pten*_v3_33198_1_1) so these seem like excellent candidates that could ultimately allow testing whether beyond *Smed* there is the negative relationship between Wnt/sexual reproduction versus head regeneration as proposed here. However, at present, given the strong likelihood from *Smeds* that the *ophis*-dependent beta-catenin portion does not influence head regeneration at all, the burden of proof rests on the authors to show functional support for the idea that Wnt levels from the sexual reproduction system participate can interfere with head regeneration in any flatworm species.

Thanks again for an insightful suggestion. We focused on the *ophis* homologue of *Cura* sp. as the regeneration-deficient species closest to *Smed* and the only species we had sufficient animal numbers available. Specifically, we fed *Cura* hatchlings with control and *ophis*(RNAi) past the sexual maturation point for a total of 14 RNAi-feeds. However, even though RNAi achieved a significant knock-down as shown by qPCR, all animals developed a gonopore and all major RS components and even started laying eggs (see A and B below). Even though *ophis* mRNA was only reduced by ~50% in this experiment (B), the results match a previous exploratory experiment with > 90% mRNA depletion and likewise maintained egg production. We therefore believe that *ophis* may very well be another example of a gene function that is not deeply conserved in planarian phylogeny. However, since we cannot rule out insufficient RNAi penetrance or functional redundancy with other GPCRs and since the experiment neither proves nor disproves our model (see above), we opted not to include these experiments in the publication.

A

B

152. Based on data presented here, there is an ambiguity in the interpretation of whole tissue or whole animal beta-catenin levels that affects the tests of the main hypothesis. How much of the beta-catenin measured here is relevant for head/tail regeneration? As the authors nicely show, several Wnt pathway components including feedback regulators such as axin and notum, have expression in the reproductive system, and therefore reproductive system components are likely themselves sources of local Wnt pathway activation. Additionally, sexual Smeds undergo head regeneration seemingly equally as well as the asexual Smeds. Therefore, the simplest interpretation of the western blotting results in Fig 5D is that a substantial portion of the detected animal-wide beta-catenin gradient in sexual animals is likely to be due to beta-catenin protein acting locally within the reproductive system, rather than as suggested that sexual animals have an intrinsically higher gradient of beta-catenin that is actively functional in head-tail determination (for example, in muscle). This first more likely conclusion would also affect the interpretation of western blotting measurements of whole-animal levels of beta-catenin in non-regenerative species. If much of the beta-catenin measured across species is simply a readout of amounts of reproductive tissue present in animals, it is less clearly a relevant or separate metric for predicting regeneration ability according to the overall hypothesis. This issue affects Fig 6g-i and the analysis of beta-catenin levels across species, because from Figure 5d, at least ~2/3 of the beta-catenin protein signal from tails of the sexual Smeds is lost in ophis RNAi and therefore not relevant to head-tail fating (because ophis inhibited animals have not been noted to have problems with head/tail identity). Because the fraction of beta-catenin used for tail patterning might differ across species, there is a concern that western blotting might not give sufficiently resolved information to weigh in strongly on the hypothesis.

We fully agree with the reviewer's analysis of the limitations of our current Western blotting assay.

As already detailed in our rebuttal of point 1), our model explicitly considers the possibility of significant but non-regeneration-relevant Wnt signalling in the RS. We also agree that the same caveat applies to the comparative analysis of Wnt signalling levels in Fig. 5 and 7, so we consistently refer to the assay as "proxy" for Wnt signalling.

We also state explicitly in the discussion that "Above all, our working model stresses the need for a more fine-grained understanding of the cellular sources and targets of Wnt signalling, both during head regeneration and in the formation and maintenance of the reproductive system."

Bottom line: Yes, the assay isn't perfect, but it is the only one we have available, and its limitations are clearly stated in the text.

3. I must say that the manuscript really inadequately references the very relevant and abundant scope of literature revealing the roles for Wnt signaling in head and tail regeneration in planarians. The authors sufficiently reference their own work but unfortunately not important work from the rest of the field that also lays the foundation leading to this study and helps to interpret its findings. For example, many studies not cited here have observed suppression of headless phenotypes by beta-catenin RNAi, which are relevant for interpreting the results from similar approaches used here across flatworm species. Many other studies not cited here have reported on the mechanism of injury-induced Wnt activation or inhibition as well as uses for systemic Wnt components in control of head/tail pattern which would be relevant for considering the possibly many ways that species could evolve higher Wnt signaling (which signals and at what steps) that might block head regeneration. Other studies have also examined beta-catenin and the reproductive system, with one study even showing beta-catenin immunostaining in the reproductive system of *S. polychroa* (Sureda-Gómez 2016). The neglect of abundant literature in the field gives the unfortunate impression that the authors have not deeply considered their study in light of other findings.

We can assure the reviewer that we remain on top of the planarian regeneration literature and that we critically assess our data in light of published data. First, a detailed consideration of the many genes and pathways that have been implicated in planarian regeneration is beyond the broad scope of this study and the journal's restrictive number of allowed references (50...).

Second, we explicitly discuss the possibility of regeneration-specific mechanisms as evolutionary drivers of regeneration defects. Specifically, the text reads:

“Similarly, the model assumes a functional excess of steady-state Wnt signalling as a common cause of regeneration defects. Though again plausible on the basis of the *β-Catenin-1(RNAi)*-induced rescue of head regeneration defects across planarian phylogeny, an alternative interpretation is that Wnt inhibition constitutes a deep developmental constraint in the neoblast-mediated formation of the planarian head. Accordingly, Wnt inhibition might be able to override all upstream physiological control mechanisms, inclusive of potentially Wnt-independent causes of head regeneration defects, and thus bypass, rather than rescue head regeneration defects”.

However, we do agree that this statement and therefore include the following citations:

Tewari et al., 2018; PMID: 30485821

Gavino et al, 2013; PMID: 24040508

Roberts-Galbraith et al., 2013 PMID: 23297191

Third, we are well aware of the Sureda-Gómez study but also of the scant detail on important controls in this study (e.g., a consideration of the remaining signal in *bCatenin-1(RNAi)* and a general evaluation of the degree of background staining). However, we agree that the *nanos* in situ data in the supplemental figure is unaffected by these concerns and therefore included the citation in the text. The section now reads:

“This finding is consistent with previous findings in the *Smed* sister species *S. polychroa*⁶⁷ and with the expression of several Wnt receptors and other pathway components in multiple reproductive system-associated tissues (Fig. 5e, S. Fig. 5b,c).”

4. The monoclonal antibodies described here, both the antibody newly reported for yolk proteins and the antibody described in 2017 for beta-catenin, should be deposited to a commercial or public hybridoma bank. This would ensure the long-term durability of the reagents for the field and also critically to meet rigor and reproducibility standards that allow findings of the work to be independently verified by other labs.

Thank you for the suggestion. We haven't pursued this option so far, as the antibodies are publicly available via the MPI-CBG antibody facility upon request. We have added a sentence to the material and methods, which now read: “Antibodies are available upon request (via the MPI-CBG antibody facility”).

5. More information about production of the EO95 antibody is needed in the methods, rather than citing prior work with the antibody core facility.

Thank you, and done. We have expanded the method section to include more detailed information on the EO95 production. The text now reads: “The FERRITIN antibody EO95 was raised against a soluble protein fragment following an established pipeline at the MPI-CBG MPI-CBG Antibody Facility¹⁷. In short, two-day-old egg capsules laid at 20°C were homogenised in 700 µl cold RIPA buffer (150 mM NaCl, 50 mM Tris-Hcl pH 8.1, 0.1% SDS (w/v), 1% Sodium Deoxycholate, 1% NP-40) and then incubated for 10 minutes at 95°C. Proteins were separated by SDS-Page and stained with KCl. The band containing FERRITIN was cut out and eluted by electro-elution into migration buffer (pH 8.5) containing 25 mM Tris, 0.19 M Glycine, and 0.1 % SDS. The procedure was repeated until a protein concentration of 1.6 mg/mL was reached. 30 µg was injected into a BALB/c mouse for immunization and 15 µg protein for boosting. Test

bleeds, as well as final antibodies, were tested via Western blot using protein lysate from asexual and sexual *Schmidtea mediterranea* as well as egg capsule lysate.”

6. Figure 6f and 6i need y-axis labels and red lines defined (are these means, medians, etc).

Done. We have labelled the y-axis and defined the red lines in the figure caption.

7. On p. 14, the authors refer to data not shown of a head-tail beta-catenin protein gradient present in most of the species investigated here. The authors either need to show the data or remove the statement, as such statements cannot be evaluated by peer review.

Done. We have removed this sentence.

8. I'm not sure that the data in Fig 5 really show a gradient of beta-catenin protein throughout the body of the asexual Smed strain. When a statistical test is used to determine which adjacent regions have more or less beta-catenin, which regions show a statistically significant difference? The overlapping error bars make it difficult to be sure, but it seems the drop between region 6 to 5 is substantial and I think likely to be significant, but for example the levels across regions 1 through 4 do not seem to be very different from each other. It seems more accurate to call this a gradient within the tail rather than an animal-wide gradient.

For a more detailed analysis of the Wnt gradient shape, please see our published manuscript (Stuckemann et al., DevCell 2017). However, we have generally revised the text to de-emphasise the shape of the Wnt signalling profile for better alignment with the more coarse-grained analysis in this study.

9. It is interesting that the beta-catenin and APC RNAi was not successful in *P. tenuis*. Is it possible that the particular dsRNAs chosen for these experiments might simply be ineffective, or that the target proteins have a particularly long half-life, or tissues themselves undergo very slow turnover etc, rather than indicating a general lack of RNAi in that organism? I agree it is suggestive of lack of useful RNAi in general for that organism, but the claim is based on negative data.

In addition to the *RNAi* constructs discussed in the text, we have also targeted the single *Pten* homologues of *cdc-20* (necessary for M-phase completion and degraded thereafter; lethal *RNAi* phenotype in *Smed*) and the homologue of the anti-apoptotic *bcl-2* gene, which likewise causes rapid lethality in *Smed*. While both *dsRNAi*s were lethal in *Smed*, all fed or injected *Pten* lived happily ever after. Hence the confidence in the stated observations, and we think our wording avoids undue claims regarding the mechanisms. The respective section reads: “Interestingly, *P. tenuis* was largely refractory to *RNAi* by feeding (Fig. 4b) and even injection (not shown), thus providing a first indication that not all planarian species may be equally susceptible to systemic *RNAi*, as observed in nematodes^{59,60}.”

10. Typo p6 "iapable"

Thank you. We have changed “iapable” to “capable”.

11. The statement at the top of p.10 "these results further imply that head regeneration defects may be associated with a functional excess of Wnt pathway activity..." is a hypothesis, but it seems the authors should consider at this point the alternative of bypass suppression and present the subsequent analysis more explicitly as a test of these two possibilities. Or alternatively, that they chose to investigate this particular possibility.

Thank you for another helpful suggestion. We have reformulated this text section in order to make clear that we are addressing a hypothesis. The text now reads:

“... Previous authors have suggested that regeneration in planarians and other clades may be under selection as a necessary aspect of asexual reproduction by fission^{44,61} and, therefore, might become dispensible in egg-laying species. In this context, a positive pleiotropy of Wnt signalling in reproduction by egg-laying provides an intriguing hypothesis to explain the repeated emergence of Wnt-dependent regeneration defects in egg laying species. As a first test of this hypothesis, we examined ...”.

12. Please modify the language used to describe whether the model has been supported versus refuted. The statement describing the statistical test in beta-catenin levels reflects a bias in study design. It reads : "The statistical analysis of tail tip...indicated a tendency towards higher

20beta-catenin levels...yet without reaching statistical significance." Yet, what kind of outcome other than what was observed would have been able to disprove the hypothesized association? This text should be revised to say "but such an association was not statistically significant." There is a similar issue with the description of the results that "failed to unequivocally support" rather than, as is the case, simply failing to provide support for the hypothesis.

Thank you, and done. The respective section now reads:

“The statistical analysis of tail tip β -CATENIN-1 levels relative to reproductive strategy (fissiparous versus egg-laying) or regenerative abilities (robust versus restricted and poor) similarly indicated a trend towards higher β -CATENIN-1 levels in egg-laying and regeneration impaired species, but such tendency was not statistically significant (Fig. 6i). Although the β -CATENIN-1 quantifications in the present species panel failed to support the association between Wnt signalling levels, regeneration and reproduction, our model nevertheless remains a useful working hypothesis for guiding the future analysis of broader species samplings or cell biological analyses of Wnt signalling mechanisms in individual species. “

Similarly, I can see that the authors try to temper their conclusions about the Wnt/reproduction model in the abstract "Although initial quantitative comparisons...confirm some...they also highlight diversification..." However, based on the considerations from points 1 and 2, which predictions of the Wnt model do the authors claim are confirmed? There is not yet evidence that reproductive systems functionally impair planarian head regeneration, and as argued above there is not yet clear evidence that species with lower head regeneration ability have greater Wnt/beta-catenin signaling levels that are participating in head/tail regeneration. Perhaps it would be better to state that quantitative methods were applied to attempt to test this hypothesis but it remains unresolved.

Thank you, and done. The revised abstract now reads:

“Although quantitative comparisons amongst the collection species remained ambiguous about this hypothesis, they also uncovered significant mechanistic divergence between planarian lineages.”

Reviewer #2 (Remarks to the Author):

“...”

This work conclusively shows that increased WNT/ β -catenin signaling is the cause for reduced/failed head regeneration in multiple species. Additionally, the authors have collected ~40 species of planarians and have developed laboratory husbandry protocols which is a great resource for the community to carry out more comparative studies.

Thank you.

However, there are two main areas which need to be clarified –

1. A major portion of the work focuses on building the planarian flatworm phylogeny and predicting the ancestral state of regeneration. The analyses described in the results are obtuse and fails to build a clear picture about the evolutionary history of regeneration. It is unclear as to how the head regeneration data was used in ancestral state reconstruction. These efforts improve the phylogenetic trees for planarian flatworm but as presented, it is not certain that they provide insights on the evolution of head regeneration.

Thank you. We agree that the ancestral state reconstruction analysis was insufficiently explained in the first version of the manuscript. To address this point, we have now expanded the details in the Methods section, including explanations of how we use the regeneration data (i.e., which datasets are used in Figure 3), and modified the legend of Figure 3 accordingly. We have also added a graphical illustration of the workflow to Supplementary Figure 3 (see Fig. 3 c-e). We believe that these changes now clearly explain all technical aspects of the ancestral state reconstruction.

In terms of insights into the evolutionary history of regeneration, 1) the approach assigns quantitative probabilities of regenerative states at each node of the phylogeny based on statistical models, which represents a significant advance over previous visual inference. 2) Our work significantly expands the sampling of species within the clade, thereby increasing both the number of nodes and the quantitative experimental assessments of regeneration states. 3) We add experimental data on the regenerative abilities of a poorly studied planarian sister group (Proclitophora), which constrains the inference of regeneration states at the base of the

22lineage. 4) As a result, we obtain a defined set of regenerative state transitions that suggest, for example, frequent and bidirectional transitions between whole-body regeneration and positional regeneration deficiencies.

Overall, we believe that the above efforts significantly advance the understanding of the evolution of planarian head regeneration.

2. The tradeoff between regeneration and reproduction is one of the possibilities driving regeneration, but there can be others like predation which need to be considered and discussed as part of working models/hypotheses. Secondly, from Fig S5.a, we can observe that animals from all the three groups of head regeneration occupy both lentic and lotic habitats. This makes the correlation between mode of reproduction and habitat of the animals a bit stretched. The authors also suggest that asexual reproduction might be able to reestablish a colony at a faster pace compared to sexual animals, which would be a necessity in ephemeral ponds. However, we do not know if this is true as animals need to grow to a certain size before they can fission and the fission fragment is highly susceptible to predation. Thus, the selection pressure for regeneration can be driven by predation in the various habitats. It may be hard to measure predation levels in the wild, but I think it should be discussed in the text.

Thank you for raising this point. Although the soft and seemingly unprotected body architecture of planarians may indeed suggest that they are highly susceptible to predation, long-term studies in the British Isles suggest that planarians are preyed upon by comparatively few predators (Reynoldson 1983). In fact, Reynoldson suggests that predation on triclads may be negligible on the stony shores of lakes. In addition, planarians have abundant rhabdites in their epidermis, which provide an aspect of chemical protection/defence against predation, including, for example, the documented presence of tetrodotoxin in the body tissues and cocoons of two terrestrial planarian species of the genus *Bipalium* (Stokes et al., 2014). Therefore, the available data suggest that predation is likely to have little impact on field populations of planarians, and our model reflects this view. However, we agree that more data are needed on planarian ecology, particularly to clarify the importance of predation under different environmental conditions and, more generally, to provide a fuller picture of planarian ecology. We therefore add the following sentence to the discussion:

“...Clearly, this cannot be tested on laboratory populations and will require studies on natural populations and of their ecological context (e.g., predation).”

Questions

1. Figure 3: Phylogenetic trees are hard to read, and they are of low resolution.

Thanks and done. We have modified the terminal size font of Figure 3d-e to improve the readability of the species names.

2. *P. tenuis* was not susceptible to β -catenin(RNAi) is cool. Are there duplicate genes for β -catenin in this species?

Planarians generally have at least 4 β -catenin /Plakoglobin homologues but only one harbouring the transactivation domain that is required for TCF-mediated transcriptional activation. This is *β -catenin-1*, and we have indeed targeted the *β -catenin-1* homologue of *P. tenuis*. Besides the *β -catenin-1* and APC results in the manuscript, we further failed to observe the expected RNAi phenotypes when targeting the *P. tenuis* homologues of 2 essential cell division genes (please see response to point 1.9 above). Overall, these data indeed suggest that *P. tenuis* has a much reduced or absent RNAi response.

3. In these β -catenin(RNAi) experiments, did the animals form a head at the posterior blastema in the different species tested?

Yes. We occasionally observed ectopic head formation in the posterior blastema in *C. pinguis*, *P. torva*, and *C. robusta*, similar to what we previously showed in *D. lacteum* (Liu et al., 2013). We have not included these data in the text as we have focused on anterior regeneration here. Furthermore, we believe that the discussion adequately addresses the alternative possibility that the *β -catenin-1(RNAi)* phenotypes reflect a deep developmental constraint and therefore bypass of other upstream regulatory processes in head regeneration.

4. In Fig 4C, does the control *C. robusta* animal have 2 pharynges?

Quite possibly, yes. To be sure, histological sections or in situ hybridisations with pharynx markers would be required.

5. In Fig 4C, please indicate number of the animals with the shown phenotype rather than number of animals with no head.

Thank you, and done. We have modified the figure panel and the text of the figure caption as suggested by the reviewer.

6. Is WNT/ β -catenin signaling known to be required for yolk production in other animals? Is that a conserved function across metazoans?

Good point. To our knowledge, and based on a cursory review of the literature, there have been no reports of Wnt signalling requirements in vitellaria development in other systems. However, the platyhelminth clade Neophora (or Euneoophora), to which planarians belong, is characterised by ectolecital egg capsules in which the tiny and almost yolk-less oocytes are packaged among a large number of individual yolk cells (Laumer & Giribet 2014). The yolk cells are in turn supplied by the extensive yolk gland system of the sexually mature adults. The origin of the yolk is therefore qualitatively different from other animals in which the yolk is mostly deposited within the oocyte rather than in a separate cell type and hence also the roles of Wnt signaling in the process may not be conserved outside of the flatworms.

7. Is the expression of WNT pathway members in the reproductive system conserved across multiple planarian species?

Thank you for the suggestion. We have tried a number of in situ hybridisations in different species during the revision period. In the case of *P. torva*, we indeed see indications of Wnt signaling expression in the reproductive system (see image panel below). However, in situ hybridisations in planarians other than the model species remains a significant challenge. We find that each planarian species requires time-consuming optimisation trials, again as a likely consequence of the deep splits between planarian lineages. The images below reflect our best efforts during the review period. However, we are not yet satisfied with the *P. torva* protocol (e.g. remaining variability between specimens and patchy staining patterns) and have therefore decided not to include these data in the present manuscript. However, the comparative analysis

of Wnt pathway function in the reproductive system will certainly be an important focus of our "flagship species" approach described below.Genes potentially expressed in the reproductive system in *Planaria torva*Reviewer #3 (Remarks to the Author):

“ ... ”

Overall, the authors presented an interesting and well-elaborated study. The presentation of data is of high quality, and the topics are of broad interest to the journal readers. And the authors also proposed some working hypotheses for future studies. I mainly have comments on evolutionary scenarios (i.e., a more general scheme when considering evolution through gene evolution), signaling components, and the role of neoblasts.

Thank you for appreciating the scope and impact of our work.

Major comments

1. One major factor in probing the molecular basis of evolutionary dynamics is the gene content. Since the authors focused on canonical Wnt signaling and already have high-quality transcriptome data, it would be great to see the distribution of Wnt signaling components across planarian species. For example, is there any Wnt ligand expansion or loss in particular lineages or regeneration groups? We know that Wnt3 is absent in all examined spiralian genomes. And in some platyhelminths, Wnt6 to Wnt10, Wnt16, and WntA are all lost. Would it be possible that some Wnt ligands have not been lost in planarian species with poor regeneration? In other words, would the loss of Wnt signaling ligands be a key evolutionary transition to whole-body regeneration?

Thanks for this interesting suggestion. Up front: A comprehensive analysis of Wnt and Wnt pathway component evolution and function in planarians and flatworms in general is beyond the scope of the present study. The pioneering work of Riddiford and Olson has already highlighted the shared loss of Wnt16 and WntA in *S. mediterranea* and neodermata (Riddiford and Olson 2011), suggesting that those genes were already absent in the common ancestor of planarians and neodermata. We further failed to identify WntA homologues in other planarian species with limited regeneration ability (*Cura pinguis*, *Planaria torva* and *Camerata robusta*), suggesting that WntA is generally absent in all planarians.

However, cursory inspections of our transcriptomes indeed indicate the potential for lineage-specific changes in the Wnt gene complement, e.g. >11 Wnt transcripts in *D. lacteum* as compared to the 9 in *S. mediterranea*. However, each such case requires experimental verification to weed out transcriptome assembly artefacts and subsequent functional characterization. In general, we envisage the designation of what we call “flagship species” in each taxon in order to focus the time consuming method adaptation investments (e.g., in situ hybridization; see above) and the generation of additional sequence resources (e.g., ONP and genome sequencing/assembly) on representative species. In addition, we have started to establish a Catenulid model species, which we expect to complement the existing data on flatworm signaling pathway evolution. Bottom line: The systematic comparative characterization of Wnt pathway components and the mechanistic link to species-specific regeneration defects constitutes a significant breadth versus depth challenge that will keep us busy over the coming years.

2. A similar line to the first comment, Wnt signaling can be regulated through different ligands as well as antagonists, such as the anterior marker gene, notum. Did the author check the expression of Wnt antagonists? For example, at the transcriptome level, do some flatworms have more Wnt antagonists than others? How are these antagonists expressed in relation to their body or organ structures?

Thanks for the suggestion. We have attempted a number of in situ hybridisations in different species during the revision period. As shown below, they indicate that, for example, *Sfrp*'s are generally expressed in the head as in *S. mediterranea*, and the expression of *notum* at the anterior pole is also observed in *P. torva* and *C. pinguis* (see below). However, the in situ hybridisation protocol optimised for *S. mediterranea* is clearly sub-optimal for other planarian species and requires further time-consuming tweaking for each species. We, therefore, opted not to include these data in the present manuscript but to pursue an in-depth comparative analysis of Wnt antagonists as part of our flagship species approach and an upcoming phylum-wide signalling component overview.

3. It is well-known that Wnt signals (i.e., Wnt ligands) are carried in muscle cells. Thus, muscle cells are not only contractile but also a coordinate system with positional information. It would be interesting to see how major Wnt ligand orthologs are expressed in the flatworm collection. And how are these muscle cells correlated to the reproductive system?

Again, an interesting point. We agree and have attempted a number of Wnt related genes in situs in different species (please see figure below), but the same caveats as above apply. To our knowledge, functional cross-talk between muscles and development of the reproductive system has not been examined so far.

4. The model that the author proposed attempts to explain the tradeoff when asexual and sexual reproduction modes are under natural selection, depending on the ecological niche. I am not sure if fission (spontaneous "pinch off") and regeneration (damage and then wound responses) are the exact cellular and molecular processes. It seems that the authors consider them the same thing. For the flatworms capable of fission, do they also have wounded tissues or wound responses during fissiparous reproduction? Given wounding seems to be an initial signal for triggering damage-induced regeneration. Would it be possible to have three different reproduction modes instead of just two (i.e., damage-induced regeneration, fission, and egg-laying)?

This is indeed conceivable. However, the vast majority of planarian species show no obvious signs of head formation prior to fission (paratomy), and large animals can fission repeatedly in rapid succession at seemingly arbitrary sites. It is therefore commonly assumed that fission injury triggers regeneration, and our model reflects this premise.

However, recent data suggest that Hox genes may pre-specify zones where fission can occur in intact planarians, and RNAseq characterisations of the developing constrictions at different stages of the fission process may indeed be interesting. In practice, however, the rapid progression of fission and the small number of animals in a population undergoing fission at any one time make this a difficult undertaking and thus likely a multi-year PhD project beyond the scope of the current manuscript.

5. When reconstructing the ancestral state, did the authors consider the synapomorphy of flatworms? i.e., which clade shares the derived trait of regeneration? This is not very clear in the current manuscript writing.

We only constrain the model with the regeneration information of the terminals in the phylogenetic tree, i.e. the species. For example, *Maricola* is made up of species with limited regeneration ability. Therefore, in the tree where regeneration data is available (*C. robusta*, *P. littoralis*, *C. hastata* and *S. dioca*), we classify the *Maricola* as belonging to regeneration groups B or C, based on our data (we analysed four species) and the literature (data for three species, two of which we also analysed). However, we do not constrain the group root with any information before running the tree. To better explain the ASR and how the data were used, we have expanded the details in the Methods section, modified the legend in Figure 3 and added a graphical section in Supplementary Figure 3 (S.Fig. 3 c-e).

346. A fundamental component largely ignored in this study is the neoblast, the source pluripotent stem cells that provide all differentiating cell types. In some systems, the distribution of neoblasts along the body axis is correlated to the regeneration capabilities of particular body parts. Did the author consider this point? And how do the authors combine this consideration into their evolution model? Also, in case the species have sexual reproduction, how can the current model explain the dynamics between the behavior of somatic pluripotent stem cells and germline stem cells?

Thanks again for an astute observation. The reason why we do not consider the differential distribution of neoblasts in our model is twofold:

First, classic papers by Dubois and Wolff examined the potential differential distribution of neoblasts in the regeneration-deficient species *D. lacteum*. They concluded that it is not the differential distribution or potency of the neoblasts, but the A/P differences in the surrounding tissues that cause the regional regeneration defect (salient experiment: Transplantation of an anterior/regeneration-competent piece from an irradiated animal into the posterior/regeneration-deficient region of a recipient rescues head regeneration after amputation across the graft). Our visualisation of stem cells in *D. lacteum* (Liu et al., 2013), and occasional piwi-1 in situ or H3P staining in different planarian species during various undergraduate lab practicals (not shown) have all so far confirmed the largely uniform A/P distribution of neoblasts across the planarian phylogeny. Second, the fact that β -catenin (RNAi) can rescue head regeneration across our species panel functionally confirms the latent regenerative potential and thus the presence of neoblasts at all A/P positions across planarian phylogeny. Our model's underlying assumption of uniform neoblast distributions is therefore justified.

The relationship between neoblasts and germline stem cells remains an active area of research. However, the pioneering work of Phil Newmark has clearly shown that somatic neoblasts can 'differentiate' into germline stem cells, which in turn differentiate into the various cell types and organs of the reproductive system. The signalling mechanisms that orchestrate these differentiation events in space and time are unknown, but our results point to Wnt signalling as one of the component mechanisms. Furthermore, the germ line fate, as all but one of the lineage choices open to each newly born neoblast progeny, clearly provides a broad basis for competition between soma and germ line and thus for the trade-offs envisaged by our model.

Minor comments:

1. Figure 1: It would be easier for readers to understand different regeneration groups if the authors could color-code the dots in panel a and provide color-coded dots in panels b–d.

Figure 1 shows collection site locations and provides images of the worms used in the study. The regeneration groups are only established and defined in Figure 2. We therefore decided to leave Fig. 1 unchanged.

2. Figure 5b: It is recommended to add a tree to the clades on the left. It would also be helpful to highlight group A and fissiparous reproduction. Overall, the font size in Figure 5 is too small to read.

Done. We have inserted a tree to the left in Figure 5b, highlighting group A and the presence of fissiparous reproduction.

3. In the Introduction, please explain canonical Wnt signaling and how it is linked to beta-catenin to help readers without this background.

Done. We have inserted the sentence “A key signal in patterning the anteroposterior axis (A/P axis) is the evolutionarily conserved Wnt signalling pathway¹⁶⁻¹⁹, which signals via inhibiting the constitutive degradation of the transcriptional regulator *β-Catenin-1* and via consequent *β-Catenin-1*-dependent changes in gene expression²⁰.”

4. Replace *Smed* with *S. mediterranea* to have consistency across all flatworm species.

Thanks. We decided not to replace “*Smed*” with “*S. mediterranea*” because *Smed* is an established acronym in the field.

5. Line 76: A/P axis, spell it out (e.g., anterior/posterior axis).

Done.

6. Line 119: Write out MPI-NAT.

We have streamlined the main text. As a result, the text is considerably shorter and does not reference to the Max Planck Institute in this line.

7. Line 165: *Dugesia sicula* -> *D. sicula*

We decided to maintain "*Dugesia sicula*" instead of "*D. sicula*" since this is the first mention of the species in the text. Some readers could wonder if we are talking about *Dugesia*, *Dendrocoelum* or another genus starting with "D". Generally, we cite the complete name of the species the first time we mention them in the text.

8. Line 166: *Dugesia japonica* -> *D. japonica*

Done. We have changed *Dugesia japonica* into *D. japonica*.

9. Line 179: iapable -> incapable

Done. We have changed "iapable" to "incapable".

10. Avoid using the acronym "RS" for the "reproductive system." This adds unnecessary difficulty to the reading.

Thanks for the suggestion. We have changed the acronym "RS" to "reproductive system" in the text.

11. Line 376: *Dugesia ryukyuensis* -> *D. ryukyuensis*

We decided to maintain “*Dugesia ryukyuensis*” instead of “*D. ryukyuensis*” for the same reasons as for *D. sicula* (please see minor comment number 7).

Decision Letter, first revision:

28th June 2023

Dear Miquel,

Thank you for submitting your revised manuscript "Probing the evolutionary dynamics of whole-body regeneration within planarian flatworms" (NATECOLEVOL-221218247A). It has now been seen again by the original reviewers and their comments are below. The reviewers find that the paper has improved in revision, and therefore we'll be happy in principle to publish it in Nature Ecology & Evolution, pending minor revisions to satisfy the reviewers' final requests and to comply with our editorial and formatting guidelines.

[REDACTED]

Reviewer #1 (Remarks to the Author):

The revisions significantly improve the manuscript and I am in overall agreement for publication. The data presented in the rebuttal showing that ophis RNAi does not affect regeneration is compelling, and for this reviewer it is indeed relevant information for understanding the current nature of support for the overall hypothesis. However, the author's interpretation of this point in the rebuttal is reasonable and a good starting point for future studies. I also applaud and appreciate the authors for following up on potential conserved roles for ophis to further test the overall idea and agree other avenues will be necessary for future work. The revised text and citations have also improved the clarity and context of the work. I would still urge the authors to consider depositing monoclonal antibodies to an independent repository given that some such reagents have been lost over time previously (there are even examples from this field), and an independent storage solution would likely be useful for the long-term durability of the reagents and associated research. In sum though, I congratulate the authors on an outstanding and thought-provoking study.

Reviewer #2 (Remarks to the Author):

38The authors have meticulously addressed the concerns raised by the reviewers. We appreciate that they are sharing additional data that helps answer some of the questions raised. Ultimately, this work provides a great resource for the community to carry out comparative studies of flatworm regeneration. Additionally, the authors provide correlative data to build working hypotheses about the observed trade-offs between sexual reproduction and whole-body regeneration. We believe that this alone will be of interest to a large audience thinking about the question “Why some animals can regenerate while other cannot...”. It is our recommendation, therefore, that this work be accepted for publication with the following few minor changes -

1. Testing the head regenerative abilities across ~40 species of planarians is impressive. However, this data is hard to extract from the phylogenetic trees in Fig 3. It would be useful to tabulate the species tested and their head regeneration abilities either in a supplementary figure or as a table. This can also include the short form names that the authors have utilized in Fig 3e and other places.
2. One of the main hypotheses from the work is the trade-off between whole body regeneration and sexual reproduction. This is supported by the correlation data shown in figures 6f and 6i. We wonder whether this correlation can be strengthened by plotting the data against Šivickis groups instead of clubbed classification of Group A, and Groups B&C. The expectation is that both area occupied by yolk and, β -catenin levels drop from Šivickis group I to group V. I understand that the current Group A, B, and C classification is based on Šivickis groups, but splitting them out into 5 groups may show a clear trend.

Minor suggestions:

1. Please label the y-axes for all the plots in Fig 2b.
2. Fig 3d – please indicate that the black dot indicates bootstrap value 100.
3. Line 260 – “provides represents” – need one of those.
4. “...Clearly, this cannot be tested on laboratory populations and will require studies on natural populations and of their ecological context (eg., predation)” – the last part of this sentence is missing in the discussion.
5. Fig 6f y-axis is ‘mean area occupied by yolk’ and not ‘mean yolk content’.

Drs. **[REDACTED]** and Alejandro Sánchez Alvarado

Reviewer #3 (Remarks to the Author):

The authors have significantly revised the manuscript according to other reviewers' and my comments. They raised a reasonable argument that the in-depth analysis of the evolution of the Wnt signaling pathway is beyond the scope of the present study, which I can also understand. Nevertheless, the authors have addressed most of my concerns. Congratulations to the authors on this outstanding work.

39Our ref: NATECOLEVOL-221218247A

10th July 2023

Dear Dr. Vila-Farré,

Thank you for your patience as we've prepared the guidelines for final submission of your Nature Ecology & Evolution manuscript, "Probing the evolutionary dynamics of whole-body regeneration within planarian flatworms" (NATECOLEVOL-221218247A). Please carefully follow the step-by-step instructions provided in the attached file, and add a response in each row of the table to indicate the changes that you have made. Please also check and comment on any additional marked-up edits we have proposed within the text. Ensuring that each point is addressed will help to ensure that your revised manuscript can be swiftly handed over to our production team.

****We would like to start working on your revised paper, with all of the requested files and forms, as soon as possible (preferably within two weeks). Please get in contact with us immediately if you anticipate it taking more than two weeks to submit these revised files.****

In recognition of the time and expertise our reviewers provide to Nature Ecology & Evolution's editorial process, we would like to formally acknowledge their contribution to the external peer review of your manuscript entitled "Probing the evolutionary dynamics of whole-body regeneration within planarian flatworms". For those reviewers who give their assent, we will be publishing their names alongside the published article.

Nature Ecology & Evolution offers a Transparent Peer Review option for new original research manuscripts submitted after December 1st, 2019. As part of this initiative, we encourage our authors to support increased transparency into the peer review process by agreeing to have the reviewer

40comments, author rebuttal letters, and editorial decision letters published as a Supplementary item. When you submit your final files please clearly state in your cover letter whether or not you would like to participate in this initiative. Please note that failure to state your preference will result in delays in accepting your manuscript for publication.

Cover suggestions

As you prepare your final files we encourage you to consider whether you have any images or illustrations that may be appropriate for use on the cover of Nature Ecology & Evolution.

Nature Ecology & Evolution has now transitioned to a unified Rights Collection system which will allow our Author Services team to quickly and easily collect the rights and permissions required to publish your work. Approximately 10 days after your paper is formally accepted, you will receive an email in providing you with a link to complete the grant of rights. If your paper is eligible for Open Access, our Author Services team will also be in touch regarding any additional information that may be required to arrange payment for your article.

Please note that *Nature Ecology & Evolution* is a Transformative Journal (TJ). Authors may publish their research with us through the traditional subscription access route or make their paper immediately open access through payment of an article-processing charge (APC). Authors will not be required to make a final decision about access to their article until it has been accepted. [Find out more about Transformative Journals](https://www.springernature.com/gp/open-research/transformative-journals)

Authors may need to take specific actions to achieve [compliance with funder and institutional open access mandates](https://www.springernature.com/gp/open-research/funding/policy-compliance-faqs). If your research is supported by a funder that requires immediate open access (e.g. according to [Plan S principles](https://www.springernature.com/gp/open-research/plan-s-compliance)) then you should select the gold OA route, and we will direct you to the compliant route where possible. For authors selecting the subscription publication route, the journal's standard licensing

41terms will need to be accepted, including <https://www.nature.com/nature-portfolio/editorial-policies/self-archiving-and-license-to-publish>. Those licensing terms will supersede any other terms that the author or any third party may assert apply to any version of the manuscript.

For information regarding our different publishing models please see our <https://www.springernature.com/gp/open-research/transformativ-journals> Transformativ Journals page. If you have any questions about costs, Open Access requirements, or our legal forms, please contact ASJournals@springernature.com.

[REDACTED]

[REDACTED]

Reviewer #1:

Remarks to the Author:

The revisions significantly improve the manuscript and I am in overall agreement for publication. The data presented in the rebuttal showing that ophis RNAi does not affect regeneration is compelling, and for this reviewer it is indeed relevant information for understanding the current nature of support for the overall hypothesis. However, the author's interpretation of this point in the rebuttal is reasonable and a good starting point for future studies. I also applaud and appreciate the authors for following up on potential conserved roles for ophis to further test the overall idea and agree other avenues will be necessary for future work. The revised text and citations have also improved the clarity and context of the work. I would still urge the authors to consider depositing monoclonal antibodies to an independent repository given that some such reagents have been lost over time previously (there are even examples from this field), and an independent storage solution would likely be useful for the long-term durability of the reagents and associated research. In sum though, I congratulate the authors on an outstanding and thought-provoking study.

Reviewer #2:

Remarks to the Author:

The authors have meticulously addressed the concerns raised by the reviewers. We appreciate that they are sharing additional data that helps answer some of the questions raised. Ultimately, this work provides a great resource for the community to carry out comparative studies of flatworm regeneration. Additionally, the authors provide correlative data to build working hypotheses about the observed trade-offs between sexual reproduction and whole-body regeneration. We believe that this alone will be of interest to a large audience thinking about the question "Why some animals can regenerate while other cannot...". It is our recommendation, therefore, that this work be accepted for

42publication with the following few minor changes -

1. Testing the head regenerative abilities across ~40 species of planarians is impressive. However, this data is hard to extract from the phylogenetic trees in Fig 3. It would be useful to tabulate the species tested and their head regeneration abilities either in a supplementary figure or as a table. This can also include the short form names that the authors have utilized in Fig 3e and other places.
2. One of the main hypotheses from the work is the trade-off between whole body regeneration and sexual reproduction. This is supported by the correlation data shown in figures 6f and 6i. We wonder whether this correlation can be strengthened by plotting the data against Šivickis groups instead of clubbed classification of Group A, and Groups B&C. The expectation is that both area occupied by yolk and, β -catenin levels drop from Šivickis group I to group V. I understand that the current Group A, B, and C classification is based on Šivickis groups, but splitting them out into 5 groups may show a clear trend.

Minor suggestions:

1. Please label the y-axes for all the plots in Fig 2b.
2. Fig 3d – please indicate that the black dot indicates bootstrap value 100.
3. Line 260 – “provides represents” – need one of those.
4. “...Clearly, this cannot be tested on laboratory populations and will require studies on natural populations and of their ecological context (eg., predation)” – the last part of this sentence is missing in the discussion.
5. Fig 6f y-axis is ‘mean area occupied by yolk’ and not ‘mean yolk content’.

Drs. **[REDACTED]** and Alejandro Sánchez Alvarado

Reviewer #3:

Remarks to the Author:

The authors have significantly revised the manuscript according to other reviewers' and my comments. They raised a reasonable argument that the in-depth analysis of the evolution of the Wnt signaling pathway is beyond the scope of the present study, which I can also understand. Nevertheless, the authors have addressed most of my concerns. Congratulations to the authors on this outstanding work.

Author Rebuttal, first revision:Rebuttal

We here submit the final version of our accepted manuscript titled "Evolutionary dynamics of whole-body regeneration across planarian flatworm".

We want to thank all of our reviewers for a constructive review process, helpful suggestions, appreciation of our revision efforts and for supporting the publication of the manuscript. As detailed below, the current manuscript version incorporates the few remaining text changes suggested by the reviewers and the editorial team. We further list a number of additional text changes at the end of this document (mostly corrections of typos).

Reviewer #1 (Remarks to the author):

1. I would still urge the authors to consider depositing monoclonal antibodies to an independent repository given that some such reagents have been lost over time previously (there are even examples from this field), and an independent storage solution would likely be useful for the long-term durability of the reagents and associated research. In sum though, I congratulate the authors on an outstanding and thought-provoking study.

Thank you. We fully agree that the long term public availability of the antibodies is important, but again point out that their administration by the MPI-CBG antibody facility guarantees just that. In addition, the mixed funding model of the MPI-CBG antibody facility entails a number of legal obligations that make submission to ASCB or other repositories challenging. In the unlikely case that the CBG antibody facility should close down, these obligations will no longer apply and the clones will be passed on to ASCB or a similar repository.

Reviewer #2 (Remarks to the author):

... It is our recommendation, therefore, that this work be accepted for publication with the following few minor changes -

Thank you!

1. Testing the head regenerative abilities across ~40 species of planarians is impressive. However, this data is hard to extract from the phylogenetic trees in Fig 3. It would be useful to tabulate the species tested and their head regeneration abilities either in a supplementary figure or as a table. This can also include the short form names that the authors have utilized in Fig 3e and other places.

Thanks for this suggestion. We have prepared an additional table (Supplementary Table 1) that tabulates all the species that have been analysed in the project. The table includes the information for head regeneration (when available) and the short-form names used in the main figures.

2. One of the main hypotheses from the work is the trade-off between whole body regeneration and sexual reproduction. This is supported by the correlation data shown in figures 6f and 6i. We wonder whether this correlation can be strengthened by plotting the data against Šivickis groups instead of clubbed classification of Group A, and Groups B&C. The expectation is that both area occupied by yolk and, β -catenin levels drop from Šivickis group I to group V. I understand that the current Group A, B, and C classification is based on Šivickis groups, but splitting them out into 5 groups may show a clear trend.

Thanks for this suggestion. Unfortunately, our current species sampling insufficiently covers the Šivicki categories (e.g., *D. lacteum* as only "Group 2" representative), but this is an avenue we aim to explore further in the future. Similarly, a quantitative analysis of the strength of the regeneration defect, i.e., the relative frequency of head regeneration failure in regeneration-limited species, might strengthen the correlation between β -Catenin levels,

regeneration abilities and reproductive parameters. A thorough analysis along these lines requires larger data sets (N- numbers of regeneration assays) and also different statistical approaches, which is why we opted to defer it to a future publication.

Minor suggestions:

1. Please label the y-axes for all the plots in Fig 2b.
Done.
2. Fig 3d – please indicate that the black dot indicates bootstrap value 100.
The black dot indicated the Tricladida group. We have changed the black dot into an arrow to clarify this point.
3. Line 260 – “provides represents” – need one of those.
Thanks. We erased “provides” from the text.
4. “...Clearly, this cannot be tested on laboratory populations and will require studies on natural populations and of their ecological context (eg., predation)” – the last part of this sentence is missing in the discussion.
We have incorporated the missing part of the sentence in the text.
5. Fig 6f y-axis is ‘mean area occupied by yolk’ and not ‘mean yolk content’.
We have changed the y-axis in accordance with your comment.

Drs. Viraj Doddihal and Alejandro Sánchez Alvarado
Thank you both for your constructive review comments!

Reviewer #3 (Remarks to the author):

The authors have significantly revised the manuscript according to other reviewers' and my comments. They raised a reasonable argument that the in-depth analysis of the evolution of the Wnt signaling pathway is beyond the scope of the present study, which I can also understand. Nevertheless, the authors have addressed most of my concerns. Congratulations to the authors on this outstanding work.

Thank you!

List of additional text changes introduced by the authors in the documents:

- In “Acknowledgements”: We have introduced Oleg Timoshkin in the acknowledgements.
- In “Acknowledgements”: We have introduced the sentence “We acknowledge Instituto Florestal and the administration of the Parque Estadual do Juquery for licensing sampling and help with fieldwork (002947/2021-83)”.
- In Fig. 2d: “cf. *Spathula* sp.” changed to “cf. *Spathula* sp. 2”
- In Fig. 3d: “*Hymaniella reteunova*” changed to “*Hymanella retenuova*”.
- In Fig. 4a: We have changed the font type in the amounts of protein loaded in the gel from “italic” to “regular”.
- In Fig. 6g: We changed 50 to 55kDa
- In Fig. 5a: We have deleted the detail on the gonopore of the animal because its size was too small to be visible.

- We have substantially shorted the figure legends of Fig. 5 and 6 to conform with the journal's formatting requirements.Final Decision Letter:

14th September 2023

Dear Miquel,

We are pleased to inform you that your Article entitled "Evolutionary dynamics of whole-body regeneration across planarian flatworms", has now been accepted for publication in Nature Ecology & Evolution.

Over the next few weeks, your paper will be copyedited to ensure that it conforms to Nature Ecology and Evolution style. Once your paper is typeset, you will receive an email with a link to choose the appropriate publishing options for your paper and our Author Services team will be in touch regarding any additional information that may be required

Due to the importance of these deadlines, we ask you please us know now whether you will be difficult to contact over the next month. If this is the case, we ask you provide us with the contact information (email, phone and fax) of someone who will be able to check the proofs on your behalf, and who will be available to address any last-minute problems . Once your paper has been scheduled for online publication, the Nature press office will be in touch to confirm the details.

Acceptance of your manuscript is conditional on all authors' agreement with our publication policies (see www.nature.com/authors/policies/index.html). In particular your manuscript must not be published elsewhere and there must be no announcement of the work to any media outlet until the publication date (the day on which it is uploaded onto our web site).

Please note that *Nature Ecology & Evolution* is a Transformative Journal (TJ). Authors may publish their research with us through the traditional subscription access route or make their paper immediately open access through payment of an article-processing charge (APC). Authors will not be required to make a final decision about access to their article until it has been accepted. [Find out more about Transformative Journals](https://www.springernature.com/gp/open-research/transformative-journals)

Authors may need to take specific actions to achieve [a](https://www.springernature.com/gp/open-research/funding/policy-compliance-)

48faqs"> compliance with funder and institutional open access mandates. If your research is supported by a funder that requires immediate open access (e.g. according to Plan S principles) then you should select the gold OA route, and we will direct you to the compliant route where possible. For authors selecting the subscription publication route, the journal's standard licensing terms will need to be accepted, including https://www.nature.com/nature-portfolio/editorial-policies/self-archiving-and-license-to-publish. Those licensing terms will supersede any other terms that the author or any third party may assert apply to any version of the manuscript.

An online order form for reprints of your paper is available at https://www.nature.com/reprints/author-reprints.html. All co-authors, authors' institutions and authors' funding agencies can order reprints using the form appropriate to their geographical region.

We welcome the submission of potential cover material (including a short caption of around 40 words) related to your manuscript; suggestions should be sent to Nature Ecology & Evolution as electronic files (the image should be 300 dpi at 210 x 297 mm in either TIFF or JPEG format). Please note that such pictures should be selected more for their aesthetic appeal than for their scientific content, and that colour images work better than black and white or grayscale images. Please do not try to design a cover with the Nature Ecology & Evolution logo etc., and please do not submit composites of images related to your work. I am sure you will understand that we cannot make any promise as to whether any of your suggestions might be selected for the cover of the journal.

You can generate the link yourself when you receive your article DOI by entering it here: http://authors.springernature.com/share.

[REDACTED]

P.S. Click on the following link if you would like to recommend Nature Ecology & Evolution to your librarian <http://www.nature.com/subscriptions/recommend.html#forms>

** Visit the Springer Nature Editorial and Publishing website at http://editorial-jobs.springernature.com?utm_source=ejp_NEcoE_email&utm_medium=ejp_NEcoE_email&utm_campaign=ejp_NEcoE for more information about our career opportunities. If you have any questions please click [here](mailto:editorial.publishing.jobs@springernature.com).**